# Compositional Reasoning with Transformers, RNNs, and Chain of Thought

**Gilad Yehudai**
Courant Institute of Mathematical Sciences
New York University
gy2209@nyu.edu

**Noah Amsel**
Courant Institute of Mathematical Sciences
New York University
noah.amsel@nyu.edu

**Joan Bruna**
Courant Institute of Mathematical Sciences,
& Center for Data Science, New York University
Center for Computational Mathematics, Flatiron Institute
jb4496@nyu.edu

## Abstract

It is well understood that different neural network architectures are suited to different tasks, but is there always a single best architecture for a given task? We compare the expressive power of transformers, RNNs, and transformers with chain of thought tokens on a simple and natural class of tasks we term Compositional Reasoning Questions (CRQ). This family captures multi-step problems with tree-like compositional structure, such as evaluating Boolean formulas. We prove that under standard hardness assumptions, *none* of these three architectures is capable of solving CRQs unless some hyperparameter (depth, embedding dimension, and number of chain of thought tokens, respectively) grows with the size of the input. We then provide constructions for solving CRQs with each architecture. For transformers, our construction uses depth that is logarithmic in the problem size. For RNNs, logarithmic embedding dimension is necessary and sufficient, so long as the inputs are provided in a certain order. For transformers with chain of thought, our construction uses $n$ CoT tokens for input size $n$. These results show that, while CRQs are inherently hard, there are several different ways for language models to overcome this hardness. Even for a single class of problems, each architecture has strengths and weaknesses, and none is strictly better than the others.

## 1 Introduction

Large language models [Touvron et al., 2023, Anil et al., 2023, Achiam et al., 2023] are increasingly used to perform logical reasoning and other problems that require algorithmic thinking. To understand the power and limitations of these models, it is essential to determine what kinds of computational problems they are capable of solving. To this end, a long line of theoretical work has studied the expressive power of various language modeling architectures using simple tasks like copying strings, recognizing formal languages, and determining if a graph is connected [Jelassi et al., 2024, Sanford et al., 2024b, Strobl et al., 2024]. Such studies typically provide constructions or prove impossibility results for a particular architecture or paradigm—such as recurrent neural networks, transformers, and transformers with test-time scaling ("chain of thought")—following the progress of the field.[1]

---

[1]In this paper, we consider transformers with and without chain of thought to be distinct architectures. We also distinguish between "shallow" (constant number of layers compared to the input size) and "deep" transformers. While these are all types of transformer, our results show that they are qualitatively very different.

39th Conference on Neural Information Processing Systems (NeurIPS 2025).

Table 1: Comparison between our methods for solving CRQs on $n$ nodes. Each architecture minimizes one kind of resource at the expense of the others. For ease of comparison, we assume the depth of the CRQ tree is $\log n$. By parallel runtime, we mean the runtime using an unlimited number of parallel processors.

| Architecture | Num. Parameters | Runtime | Parallel Runtime |
|---|---|---|---|
| Deep transformer (Section 4) | $O(L \log^2 n)$ | $O(L\, n^2 \log n)$ | $O(L)$ |
| RNN (Section 5) | $[O(\log n), O(n)]^2$ | $[O(n \log n), O(n^2)]$ | $O(n)$ |
| Chain of Thought (Section 6) | $O(\log^2 n)$ | $O(n^2 \log^2 n)$ | $O(n)$ |

In this paper, we take a broader view. We compare the abilities of several different language modeling architectures to solve a class of problems that we call Compositional Reasoning Questions (CRQs). While it is intuitive that each architecture has strengths and weaknesses, a rigorous understanding of this phenomenon is incomplete. Theoretical comparisons between architectures often use tasks that are selected to make one model look good. For instance, copying an arbitrarily long string is impossible for an RNN with fixed memory, but easy for a transformer [Jelassi et al., 2024]. In this work, we prove that the relationships between model classes can be much subtler and more interesting. Our study of CRQs prove that there are unavoidable trade-offs between the architectures we consider, even for a single problem. None of these architectures can solve CRQs without a strong dependence on the size of the input. However, each architecture allows us to minimize the use of some computational resource compared to the others (see Table 1). CRQs are simultaneously hard in different ways for different architectures, but not intractable for any one of them. Thus, the CRQ task provides a crisp and formal way to characterize the *essential* differences between these architectures.

Our motivation for defining and studying Compositional Reasoning Questions is as follows. Many multi-step reasoning tasks share a common tree-like structure, meaning that they are hierarchical compositions of smaller tasks. Consider the following examples:

- *What is $(6 + 2) \cdot (4 - 5)$?*
- *Among all U.S. states, which one's highest mountain has the lowest elevation?*

To answer these questions, we must first solve several sub-tasks, like "*What is $6 + 2$?*" and "*What is the elevation of the highest point in Hawaii?*" While some of these sub-tasks can be solved in parallel, the overall task cannot be solved without first gathering the answers to the sub-tasks. Furthermore, sub-tasks may be nested hierarchically to form more complex tree structures, as in $5 \cdot (8 - (9 \div (2 - 1)))$. Each sub-task corresponds to a non-leaf node of the tree, and the answer to the overall questions corresponds to the root node. See Figure 1 for an example tree. The paradigmatic example of compositional reasoning is Boolean formula evaluation, which is the problem of determining the truth-value of an expression like $(\mathtt{T} \vee \mathtt{F}) \wedge (\neg(\mathtt{F} \wedge \mathtt{T}))$. This is a problem of fundamental importance in basic logic and in complexity theory, where it is one of the key $NC^1$-complete problems, but it has been surprisingly understudied in prior work on LLM reasoning. Various other tasks studied in the literature also share this hierarchical structure [Sinha et al., 2019, Feng et al., 2024]. Compositional Reasoning Questions, which we define in Section 3.1, provide a simple and unified framework for studying reasoning tasks that exhibit compositionality. In particular, we prove that the CRQ task captures Boolean formula evaluation. Therefore, we believe that CRQs are of fundamental importance as a yardstick for LLM reasoning.

We study the abilities of deep transformers, recurrent neural networks, and shallow transformers with chain of thought to solve these problems. Our main contributions are as follows:

1. In Section 3.1, we present Compositional Reasoning Questions, a simple, formal framework based on semantic similarity for studying LLM reasoning on arbitrary tree structures.

2. In Section 4, we prove that transformers with constant depth cannot solve arbitrary CRQs (Theorem 4.3), but transformers with depth $L$ can solve all CRQs of depth up to $L$ (Theorem 4.1).

3. In Section 5, we prove that RNNs with constant hidden dimension cannot solve arbitrary CRQs (Theorem 5.5), but RNNs with $O(\log n)$ hidden dimension and constant depth can

---

[2]Depending on the ordering of the inputs, see more details in Section 5.

solve all CRQs of size $n$ (Theorem 5.4). This ability depends on the inputs being arranged in a particular order (Algorithm 1); if they are ordered adversarially, RNNs require $O(n)$ hidden dimension (Theorem 5.2).

4. In Section 6, we prove that transformers augmented with $O(\log n)$ CoT tokens cannot solve CRQs of size $n$, but transformers augmented with $O(n)$ CoT tokens can (Theorem 6.1).

Each of these results fills a gap in the literature; taken together, they demonstrate the fundamental trade-offs between different models (Table 1). While deep transformers are highly parallelizable, they require $O(\log n)$ depth in the worst case, so the model size must depend (albeit mildly) on the problem size. Likewise, RNNs use little compute but must grow to handle larger problems, although their success depends on the order of the inputs. Chain of thought allows a single, logarithmic-size model to handle any CRQ, but it runs slowly and is not parallelizable. Overall, our work reveals a rich complexity landscape for an important class of reasoning problems.

## 2   Related Work

**Expressive power of transformers.** Our work belongs to a large body of research studying the representational capacity of transformers, chain of thought prompting and RNNs. Transformers with unbounded depth are known to be universal approximators [Yun et al., 2019], and can simulate Turing machines [Wei et al., 2022a, Merrill and Sabharwal, 2023b] if their size can grow with the sequence length. Looped transformers, which repeat a fixed sequence of transformer layers a variable number of times depending on the length of the input, can implement a simple but universal programming language [Giannou et al., 2023]. Several works studied the expressive power of shallow transformers in solving certain representative tasks. Sanford et al. [2024c] study problems like sparse averaging and matching, Yehudai et al. [2024] study counting, and Amsel et al. [2025] study nearest neighbor. A long line of work [Hahn, 2020, Hao et al., 2022, Merrill et al., 2022, Strobl et al., 2024] has used circuit complexity classes to

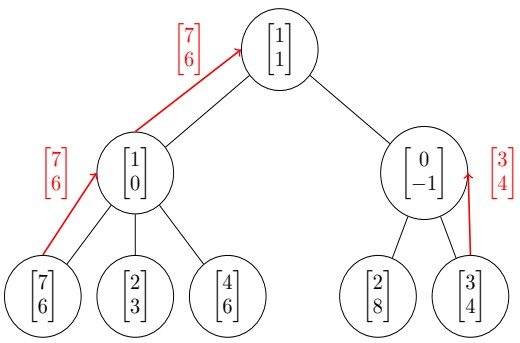

Figure 1: Example of a compositional reasoning question. Inside each node is the vector corresponding to this node. Red lines indicate the answer of each sub-question, given by $\arg\max_{u \in \mathcal{C}(v)} \{\mathbf{x}_v, \mathbf{x}_u\}$, where $\mathcal{C}(v)$ are the children of node $v$.

characterize the power and limitations of transformers. Merrill and Sabharwal [2023c] show that constant-depth transformers can only compute functions that are computable by Boolean circuits in the complexity class $\mathsf{TC}^0$. We use this result to prove that they cannot solve CRQs either (Theorem 4.3). Finally, Chen et al. [2024] study decoder-only transformers, in which each token can attend only to the previous tokens in the sequence, and prove that $L$-layer decoders are strictly weaker than $L + 1$-layer decoders for the task of function composition.

Past work has also studied the expressive power of transformers with logarithmic-depth. [Sanford et al., 2024a,b] focus on graph tasks, showing how the parallelism of transformers gives them an advantage over other architectures. Liu et al. [2022] show the ability of logarithmic-depth transformers to simulate finite automata, and Merrill and Sabharwal [2025] extend these results by weakening their assumptions. We note that as implied by our Lemma 4.2, CRQs in general cannot be solved by finite-state automata.

**Power of RNNs** An older line of research studied the capacities of recurrent networks, usually by relating them to automata [Merrill, 2019, Korsky and Berwick, 2019, Merrill et al., 2020]. However, they generally assume that the order of the inputs is fixed and that the size of the network is constant in the size of the input. We study models whose hyperparameters (size, number of CoT tokens) change with the problem size, and we study the effect of both benevolent and adversarial orderings of the inputs.

**Chain of Thought prompting.** Chain of thought (CoT) [Reynolds and McDonell, 2021, Wei et al., 2022b, Nye et al., 2021] is a method that enhances the ability of transformers to solve logical reasoning tasks. Rather than producing the answer all at once, the model is allowed to autoregressively generate a series of intermediate tokens to help it carry out each step in the solution. Li et al. [2024] show that constant-depth transformers that generate $T$ CoT tokens can simulate Boolean circuits of size $T$. Merrill and Sabharwal [2023b] prove they can simulate $O(T)$ steps of a Turing machine.

Most closely related to our work is that of Feng et al. [2024], which studies the power of chain of thought in solving arithmetic problems. Like CRQs, arithmetic expressions correspond to tree of subproblems and are solved by working up the tree. Unlike their work, in which the tree structure is indicated by parentheses in the sequence, we encode the tree structure into the positional encodings of the tokens. However, both problems are $\mathsf{NC}^1$-hard. Like Feng et al. [2024], we prove that our task is solvable by a constant depth transformer using chain of thought, but not otherwise. However, there are several differences: (i) We provide an explicit construction for solving CRQs using *logarithmic* depth transformers without chain of thought. (ii) While their task is unsolvable by RNNs with a hidden dimension of $o(n/\log n)$, we provide an explicit construction of an RNN that solves CRQs using only $O(\log n)$, but does not work for all orderings of the input. (iii) Our chain of thought solution generates $n$ tokens, while theirs requires $\Omega(n^2)$ tokens. Such different conclusions emphasize the importance of the input format; in particular our more favorable scalings hint at the importance of leveraging prior information about the hierarchical structure. In Appendix F we generalize our definition of CRQ to also capture arithmetic operations, as studied in Feng et al. [2024].

## 3 Problem Formulation and Preliminaries

**Notation.** We use bold letters for vectors, e.g. $\mathbf{x}, \mathbf{y}$. Let $(\mathbf{x})_{i:j} \in \mathbb{R}^{j-i+1}$ denote entries $i$ through $j$ of $\mathbf{x}$, and $(\mathbf{x})_{-1}$ its last entry. For $n \in \mathbb{N}, [n] = \{1, \ldots, n\}$. Let $T = (V, E)$ be a rooted tree with root $v_r \in V$. The *depth of a node* $v \in V$ is the length of the unique path from $v \in V$ to $v_r$. The *depth of the tree* is the largest depth of its nodes. The *parent* of a node $v \in V$ is the node connected to $v$ on the path to the root and denoted by $\mathcal{P}(v)$. The *children* of $v$ are the nodes whose parent is $v$, and are denoted by $\mathcal{C}(v)$. The *degree* of a node is the number of its children. Thus, leaves have a degree of 0. We say that a node $u$ is a *descendant* of $v$ if there is $i$ such that $\mathcal{P}^{(i)}(u) = v$, where $\mathcal{P}^{(i)}$ is the parent function composed with itself $i$ times. We also say in this case that $v$ is an *ancestor* of $u$.

### 3.1 Compositional Reasoning Questions

We formally define the class of Compositional Reasoning Questions as follows:

**Definition 3.1.** *A **Compositional Reasoning Question (CRQ)** is a rooted tree $T = (V, E)$ with root $v_r \in V$, where each node in the tree $v \in V$ is labeled by a vector $\mathbf{x}_v \in \Gamma^d$. Here $\Gamma \subset \mathbb{R}$ is some finite vocabulary of constant size and $d \in \mathbb{N}$.*

*The **size** of the CRQ is defined as $|V|$. The **answer** to the CRQ, denoted as $\mathcal{A}(v_r)$, is defined recursively: For a leaf $u \in V$ we define $\mathcal{A}(u) = \mathbf{x}_u$. For a non-leaf node $v \in V$ we define:*

$$\mathcal{A}(v) = \arg\max_{\mathbf{x}_u, u \in \mathcal{C}(v)} \langle \mathcal{A}(u), \mathbf{x}_v \rangle \ .$$

*We refer to every node $v \in V$ which is not a leaf as a **sub-question**.*

In Appendix A.2, we prove that Boolean formula evaluation is a special case of CRQ. Intuitively, the leaves represent Boolean literals (true and false), the sub-questions represent logical operations (and, or, etc.), the tree topology encodes the order of operations, and the answer is the truth value of the overall expression. Using vector representations is not only more general, but also makes for simple constructions using language modeling architectures like attention.

In the above definition we assume that the CRQs are defined over some finite vocabulary $\Gamma$. We can think of it as $\Gamma = \{0, \pm 1, \ldots, \pm 9\}$ for simplicity, although any other vocabulary is allowed too.

CRQs are thus inherently hierarchical tasks, where the usual sequential structure is replaced by a tree structure. Therefore, we will consider learning CRQs with models that can leverage this tree structure. Each node $\mathbf{v}$ is embedded as the vector $(\mathbf{x}_v, \mathbf{z}_v, \mathbf{z}_{\mathcal{P}(v)}, \ell(v))$, where $\mathbf{x}_v$ is the label, $\mathbf{z}_v$ is a positional embedding, and $\ell(v)$ is the depth of $v$ in the tree.

## 3.2 Transformers

We now provide a formal definition of the transformer architecture that will be used throughout the paper. The input sequence is $\mathbf{x}_1, \ldots, \mathbf{x}_n \in \mathbb{R}^d$. We concatenate **Positional Encodings (PE)** $\mathbf{e}_1, \ldots, \mathbf{e}_n \in \mathbb{R}^e$ to the inputs. The positional encodings cannot depend on the values of the vectors $\mathbf{x}_i$, only on their positions (namely, on $i$). The input to the transformer is the sequence $\tilde{\mathbf{x}}_i = \begin{pmatrix} \mathbf{x}_i \\ \mathbf{e}_i \end{pmatrix} \in \mathbb{R}^{d+e}$. The transformer backbone contains alternating layers of hardmax, single-head self-attention and feed forward networks with skip connections. For simplicity, we do not use layer normalization or attention masking (though the lower bounds like Theorem 4.3 generally hold against models that include them). Formally, let $\boldsymbol{h}_i^{(\ell)}$ denote the hidden embedding output after $\ell$ transformer blocks corresponding to the $i$th input. Define:

$$\boldsymbol{h}_i^{(0)} = \tilde{\mathbf{x}}_i \tag{1}$$

$$\boldsymbol{h}_i^{(\ell+1/2)} = \boldsymbol{h}_i^{(\ell)} + \boldsymbol{V}_\ell \underset{\boldsymbol{h} \in \{\boldsymbol{h}_1^{(\ell)}, \ldots, \boldsymbol{h}_n^{(\ell)}\}}{\arg\max} \left( \boldsymbol{h}^\top \boldsymbol{K}_\ell^\top \boldsymbol{Q}_\ell \boldsymbol{h}_i^{(\ell)} \right) \tag{2}$$

$$\boldsymbol{h}_i^{(\ell+1)} = \mathrm{MLP}_{\theta_\ell} \left( \boldsymbol{h}_i^{(\ell+1/2)} \right) \tag{3}$$

where $\boldsymbol{V}_\ell, \boldsymbol{K}_\ell, \boldsymbol{Q}_\ell \in \mathbb{R}^{(d+e) \times (d+e)}$ and $\theta_\ell$ are weight matrices and $\sigma(\cdot)$ is the ReLU function. Ties in the $\arg\max$ are broken arbitrarily. Finally, an unembedding layer $\mathbf{y}_i = \begin{pmatrix} \boldsymbol{I}_d & \cdot \\ \cdot & \cdot \end{pmatrix} \boldsymbol{h}_i^{(L)}$ discards entries corresponding to the positional encoding. We consider $\mathbf{y}_n$ to be the output of the model. Following previous work [Merrill and Sabharwal, 2023a,c], we assume that the inputs, weights, and intermediate representations of the network can all be represented using $O(\log n)$ bits of precision. We use hardmax attention for ease of analysis, following previous works on the theory of transformers. However, all our constructions can be extended to softmax. To do so, we would modify the statements of Theorems 4.1 and 6.1 to say that our constructions approximate the target function up to arbitrary error $\epsilon > 0$ that is independent of $n$, rather than exactly expressing the target. The proofs would then tune the temperature of the softmax function so that it is sufficiently close to hardmax to achieve approximation error $\epsilon$.

## 4 Depth in Transformers is Necessary and Sufficient

In this section we provide two complementary results about the power of transformers to solve CRQs. We first show that deep transformers can solve any CRQ so long as the depth of the transformer is at least the depth of the CRQ tree. We then prove a conditional lower bound showing that constant depth transformers cannot solve all CRQs. Combining both results, we conclude that depth is both necessary and sufficient for transformers to solve CRQs.

### 4.1 Deep transformers can solve CRQs

**Theorem 4.1.** *For any $L, n \in \mathbb{N}$ there exists a transformer $T$ with depth $L - 1$ and embedding dimension $O(d + \log(n))$ that solves all CRQs with at most $n$ nodes and depth at most $L$.*

The proof appears in Appendix A.1. First, note that the depth of the transformer depends only on the depth of the tree of the CRQ. This is one of the main strengths of transformers: parallelism. Namely, the transformer is able to solve all the sub-questions in the tree of the same depth with just one layer.

We also emphasize that it is probably not possible to solve CRQs of size $n$ with less than $\log(n)$ precision. The reason is that a transformer without positional encoding is invariant to the order of the tokens. However, the CRQ has an inherent structure, a tree, which without it it cannot be solved. Thus, just to be able to provide even the simplest positional encoding which is numbering the nodes of the tree from $1$ until $n$ requires $O(\log(n))$ bits (cf. Merrill and Sabharwal [2023a]). Our construction uses a more complex positional encoding that captures the tree structure while still using only $O(\log(n))$ bits.

We now give a short intuition for our construction. We first define a positional encoding vector for each node that has four parts. For each non-leaf node $v$, we define an identifying vector $\mathbf{z}_v \in \{\pm 1\}^{O(\log(n))}$

such that $\langle \mathbf{z}_v, \mathbf{z}_u \rangle \ll \|\mathbf{z}_v\|^2$ for any nodes $u, v \in V$. For leaf nodes, we let $\mathbf{z}_v = \mathbf{0}_{O(\log(n))}$. This is the first part. The second part of node $v$'s positional encoding vector is the identifier of its parent, $\mathbf{z}_{\mathcal{P}(v)}$. The third part is the depth of $v$ in the tree. The fourth is an indicator variable that is 1 when the depth is $L - 1$, where $L$ is the depth of the tree, and 0 otherwise. These four parts are all concatenated to the value of the node $\mathbf{x}_v$.

We construct each layer of the transformer to be exactly the same. Using the positional encoding, we construct the following attention pattern:

- Each non-leaf tokens that have depth smaller than $L - 1$ attends only to itself.
- Each non-leaf tokens that has depth $L - 1$ attends only to its children. Specifically, it attends to the child child that answers its sub-question, one whose value vector has the largest correlation with the non-leaf token.
- Leaf tokens can do as they like, as they will not be attended to in succeeding levels.

By constructing this attention pattern, all the sub-questions in layer $L - 1$ are solved simultaneously. We then use the MLP to reorganize the tokens. We increase the counter of the depth of every token, and update the indicator for nodes of depth $L - 1$. Nodes that in the previous iteration had depth $L - 1$ are altered to resemble leaves, i.e. their $\mathbf{z}_v$ vector is set to zero. For these nodes, we also set their value vector $\mathbf{x}_i$ to be the answer to their sub-question, as computing in the previous attention layer. In the next attention layer, nodes previous in layer $L - 1$ will play the part of leaves and be attended to by their parents, which were previously in layer $L - 2$. Applying this construction $L - 1$ times will compute the answer to the root, which is the answer to the CRQ.

## 4.2 Constant depth transformers cannot solve CRQs

We will now show that solving CRQs using a constant size transformer cannot be done, conditional on the assumption that $\mathsf{TC}^0 \neq \mathsf{NC}^1$. Our main result in this subsection is a reduction from Binary Formula Evaluation Problem (BFEP), which is known to be $\mathsf{NC}^1$-complete (see Buss [1987]), to solving CRQ.

**Lemma 4.2.** *The CRQ problem over a finite alphabet is $\mathsf{NC}^1$-hard.*

For a full proof, see Appendix A.2. The reduction is straightforward, and resembles the reduction in Feng et al. [2024] from BFEP to arithmetic problems. The main idea is to define the vectors $\mathbf{t}_0 = \mathbf{t}_\wedge = \begin{pmatrix} 0 \\ 1 \end{pmatrix}$ and $\mathbf{t}_1 = \mathbf{t}_\vee = \begin{pmatrix} 1 \\ 0 \end{pmatrix}$ which correspond to 0 and 1 in Boolean formulas. Now note that taking argmax of the dot product with $\mathbf{t}_\wedge$ corresponds to the $\wedge$ operation, and the argmax of the dot product with $\mathbf{t}_\vee$ corresponds to the $\vee$ operation. The operation of $\neg$ is slightly more intricate and defined in Figure 3. All the operations in the reduction can be done using $\mathsf{TC}^0$ circuits, which means that the CRQ problem is $\mathsf{NC}^1$-hard.

It was shown in Merrill and Sabharwal [2023c] that transformers with constant depth, polynomial size and logarithmic bit- precision (all w.r.t $n$) are in $\mathsf{TC}^0$. Thus, the following theorem follows immediately from Lemma 4.2:

**Theorem 4.3.** *Under the assumption that $\mathsf{TC}^0 \neq \mathsf{NC}^1$, for any $L \in \mathbb{N}$ and polynomial $P(x)$, there exists $n \in \mathbb{N}$ such that no transformer with depth $L$, a number of parameters that is smaller than $P(n)$ and $O(\log(n))$ bit-precision can solve all CRQs of size $n$.*

Combining the two results of this section, we see that transformers have the power to solve the CRQ problem for each layer efficiently, however the number of layers of the transformer depends on the size of the tree. This dependence is also mandatory under the assumption that $\mathsf{TC}^0 \neq \mathsf{NC}^1$. One concrete example, which will be relevant in the next section, is a balanced binary tree with $n$ nodes, and $\log(n)$ depth. There exists a transformer that can solve all CRQs of this shape with $\log(n)$ layers, but not with a constant number of layers. In the next sections we will provide alternative approaches to solve this problem with different models using constant depth.

We note that arithmetic problems, as presented in Feng et al. [2024], can also be embedded as certain generalized CRQs. This embedding is presented in Appendix F. Thus, both Theorem 4.1 and Theorem 4.3 can be applied to arithmetic problems, if they are presented in a tree structure. It

is in general possible to apply a preprocessing procedure using a depth $O(\log(n))$ transformer to turn an arithmetic problem presented in a sequential form, into a tree structure which captures the order of arithmetic operations. Note that the above theorem readily extends to transformers with layer normalization and attention masking, since it relies on a result for Merrill and Sabharwal [2023c] that includes both elements.

## 5 Solution Using Shallow RNNs

In this section we will show that under certain assumptions, RNNs can solve CRQs. The proofs for this section appear in Appendix C. We first define RNNs in the following way:

**Definition 5.1.** *An RNN is a fully-connected neural network* $\mathcal{N} : \mathbb{R}^{d+m} \to \mathbb{R}^m$. *The inputs to the RNN are* $\mathbf{x}_1, \ldots, \mathbf{x}_n \in \mathbb{R}^d$, *and the RNN operates as* $\mathcal{N}\left(\begin{pmatrix} \mathbf{x}_i \\ \mathbf{h}_{i-1} \end{pmatrix}\right) = \mathbf{h}_i$ *where* $\mathbf{h}_0, \ldots, \mathbf{h}_n \in \mathbb{R}^m$. *The* $\mathbf{h}_i$'s *are called the **hidden states** of the RNN.* $\mathbf{h}_0$ *is defined as part of the architecture, and independent of the input data. The output of the RNN is* $\mathbf{h}_n$[3].

Note that in contrast to transformers, RNNs are not invariant to changing the order of the inputs. The reason is that the order in which the vectors $\mathbf{x}_1, \ldots, \mathbf{x}_n$ are fed to the model is part of the input itself, and not part of the architecture like positional encodings for transformers. We will show that the way the nodes of the CRQs are ordered is crucial for having an efficient solution with RNNs.

### 5.1 The order of the nodes matters

Our next result shows that there are bad orderings which force RNNs to have large hidden dimension:

**Theorem 5.2.** *Let* $T = (V, E)$ *be a balanced binary tree of size* $n$. *There exists an ordering of the nodes with the following property: Any RNN that reads the inputs in that order and solves all CRQs defined on* $T$ *must have a hidden state of size* $\Omega(n)$ *bits. That is, if the hidden state is in* $\mathbb{R}^m$ *and each entry is represented by* $p$ *bits, then* $p \cdot m = \Omega(n)$.

The proof is given in Appendix C.1. Note that Theorem 5.2 doesn't depend on the depth of the RNN. Note that it is easy to construct an RNN with a hidden dimension of size $O(n)$ that solves all CRQs. The idea is just to encode all the nodes in the hidden state, which can be done if we have enough memory, and then use a large enough neural network to solve the CRQ. Constructing such a neural network is possible by the universal approximation property [Cybenko, 1989, Leshno et al., 1993]. Since the number of possible outputs is finite (because the alphabet is finite), we can approximate the solution up to a small enough accuracy and then threshold over the output.

The proof of Theorem 5.2 uses a communication complexity argument. Specifically, we use a reduction from the set disjointness problem (see Claim C.1 for a formal definition). We then construct a communication protocol between two parties, where each one of them has knowledge of only half of the inputs. When using the bad ordering of the nodes, this communication protocol forces the first party to encode all of its inputs into the hidden state when passing it to the second party.

Feng et al. [2024] proved that a specific RNNs construction requires a hidden state of size $\Omega(n)$ to solve arithmetic problems. Our results are stronger in that: (1) Our lower bound applies to any possible construction, so long as the bad order is used, and (2) We next show that the $\Omega(n)$ memory requirement can be alleviated by re-ordering the inputs.

### 5.2 Memory-rank sort

In this subsection, we restrict ourselves to CRQs with full binary trees for simplicity; that is, the degree of each node is 0 or 2. In Appendix D, we extend the results from this section to non-binary trees, but note that every CRQ can be converted into an equivalent CRQ with a binary tree and at most twice as many notes (see Appendix D.1).

We will now introduce a sorting algorithm for trees that will allow us to solve any CRQs using an RNN with a small hidden dimension. First, we to define the memory-rank of each node:

---

[3]It is often common to define two separate outputs of the RNN, the hidden state and the prediction. Here we combine them for simplicity.

**Definition 5.3** (Memory Rank). *Let $T = (V, E)$ be a rooted tree. The **memory rank** of a node $v \in V$ is defined recursively as: $mr(v) = 0$ if $v$ is a leaf, and otherwise*

$$mr(v) = \max\left(a_{\max}, a_{\min} + 1\right)$$
$$\text{where} \quad a_{\max} = \max\left(mr(c_1(v)), mr(c_2(v))\right)$$
$$a_{\min} = \min\left(mr(c_1(v)), mr(c_2(v))\right)$$

*and $c_1(v)$ and $c_2(v)$ are the two children nodes of $v$. The memory rank of the tree $mr(T)$ is defined as the memory rank of its root.*

Intuitively, the memory rank is the smallest possible stack size needed for a stack machine to solve the CRQ if we are allowed to pick the order of the nodes. A stack machine is an automaton augmented with a stack that it can uses as memory. It processes the inputs sequentially, optionally pushing and popping a finite number of elements from its stack at each step and performing computations with the resulting vectors. We should order the inputs according to a post-ordering depth-first search of the CRQ, meaning that the position of a node in the ordering corresponds to when the depth first search *last* visits it. This ordering akin to reverse Polish notation, which notates an arithmetic expression like $(2 + 3) * 6$ as the following sequence: $2\,3 + 6\,*$. By ordering the inputs this way, we ensure that whenever the automaton reads a non-leaf node, the top two elements on the stack are the solutions to the CRQs defined by that node's two children. It can simply pop them off, compute the solution to the current node, and push the result back onto the stack to save it for later.

Even using a depth-first search ordering, the size of the stack used by this machine could grow as large as $n$ in the worst case. However, by ordering the children of each node carefully, we can reduce the stack depth to only $\log n$. To compute the CRQ corresponding to a given node $v$, we (1) solve for one of its children, (2) save this result on the stack, then (3) solve for the second child. If we solve $c_1(v)$ first, then the largest stack depth during the first stage is $mr(c_1(v))$ and the largest stack depth during the second stage is $mr(c_2(v)) + 1$. The overall largest stack depth is the maximum of these two. Thus, to use as little memory as possible, we should always start with the child whose memory rank is larger. This sorting algorithm is defined formally in Algorithm 1. See Figure 2 for an example of memory-rank sort of a tree. In Theorem 5.4, we will simulate this stack machine using a RNN with hidden dimension of size $mr(T)$.

### 5.3 Shallow RNNs can solve CRQs

We are now ready to present the main theorem of this section:

**Theorem 5.4.** *For any $n \in \mathbb{N}$ there exists an RNN with $5$ layers and hidden dimension $O(d \log(n))$ that solves any CRQ with a binary tree, if the nodes are ordered by Algorithm 1*

The proof is given in Appendix C.2. The main idea is to simulate a stack machine with the RNN. The stack will contain the vectors $\mathbf{x}_i$ that are needed for future calculations, and will have two possible operations: (1) pop out vectors, and (2) push a vector. The stack will be transferred through the RNN in the hidden state. We include the depth of each node in its positional embedding, which will help us determine which of the two operations (push or pop) to use.

At each new input we check whether the last node in the stack has a larger depth by exactly 1. If it doesn't, then we insert $v$ into the stack and move to the next input. If it does, then we pop out the last two nodes from the stack, say $u_1$ and $u_2$, and calculate the inner products $\langle \mathbf{x}_v, \mathbf{x}_{u_1} \rangle, \langle \mathbf{x}_v, \mathbf{x}_{u_2} \rangle$. We insert back to the stack either $\mathbf{x}_{u_1}$ or $\mathbf{x}_{u_2}$, whichever has a higher inner product, but with the depth of the node $v$. All the above operations can be simulated using ReLU neural networks. The memory-rank sort does the rest of the work, since it makes sure that if $v$ is the current node to be processed, the last two nodes in the stack must either be its children, or nodes with a smaller depth. Doing this recursively over all the nodes will gradually solve the CRQ.

Note that in Theorem 4.1, we also needed the embedding dimension of the transformer to be $\Omega(\log(n))$, but for a very different reason. There, the transformer is invariant to re-ordering of the input tokens, and since the output does depend on the ordering (specifically, the structure of the tree) we would need $\Omega(\log(n))$ bits just to label the tokens from 1 to $n$. For RNNs, the inputs are already ordered, and we even defined a specific ordering algorithm that aligns with our task. The reason that in Theorem 5.4 we need the size of the hidden dimension to be $\Omega(\log(n))$ is because in order to

solve CRQs, we need to store some of the inner calculations (i.e. answers to the sub-questions) while we solve others. We emphasize that this hardness is not captured by the fact that solving CRQs is $NC^1$-hard. There are other $NC^1$-hard problems that can be solved with a constant size RNN, such as the word problem on $S_5$ (see Definition 3.1 and Theorem 5.1 in Merrill et al. [2024]).

Finally, note that the memory rank of any tree with $n$ nodes is bounded by $\log(n)$ (see Lemma C.2). In fact, for a given tree structure $T$, the hidden dimension of the RNN needed to solve all CRQs with this structure is bounded by $O(d \cdot \mathbf{mr}(T))$ using our construction. We next prove that the memory-rank also provides a lower bound on the required size of the hidden dimension:

**Theorem 5.5.** *Let $n \in \mathbb{N}$, let $T$ be a rooted tree of size $n$ with some ordering of the nodes from $1$ to $n$. Suppose there exists an RNN that solves all CRQs with a tree structure of $T$ if the nodes are provided in the given ordering. Then, the hidden dimension of the RNN must have $\Omega(\mathbf{mr}(T))$ bits. In particular, an RNN that solves all CRQs on all trees of size $n$ for a given ordering must have a hidden dimension with $\Omega(\log(n))$ bits.*

The proof is given in Appendix C.3. Combining Theorem 5.4 and Theorem 5.5, we see that the memory-rank gives a complete characterization of the memory needed to solve CRQs using RNNs.

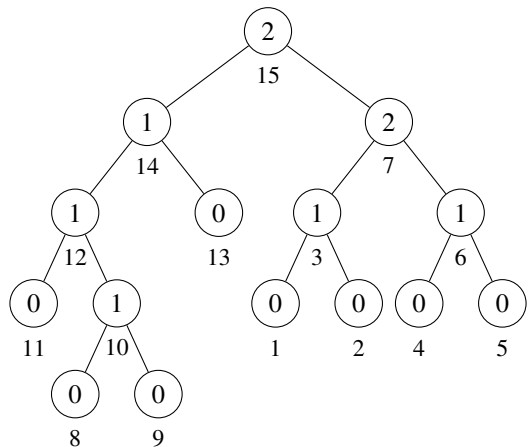

Figure 2: Memory-rank sorting of a tree. Each node is labeled with its memory rank, which ranges from 0 to 2 in this case. The number below each node is its ordering according to Algorithm 1. Note that we first traverse the right branch of the tree, since the right child of the root has a higher memory-rank than the left child.

# 6   Solution Using Shallow Transformers with Chain of Thought

In this section we show that adding the ability to produce chain of thought tokens can also help solve CRQs using a constant depth transformers. We assume in this section that the nodes are ordered in reverse BFS ordering. Namely, for a tree with $n$ nodes and depth $L$, the nodes are numbered from $1$ until $n$ starting from the nodes at depth $L$, and going up until the root which is numbered $n$. Our main result in this section is the following:

**Theorem 6.1.** *There exists a $2$-layer transformer that solves all the CRQs with trees containing $n$ nodes. The embedding dimension is $O(d + \log(n))$, the bit-precision of the transformer is $O(\log(n))$, and the number of chain-of-thought tokens generated is $n$.*

The proof idea is to solve all the sub-questions of the CRQ from the bottom-up. Each generated CoT token will be a solution to one of the sub-questions represented by a non-leaf node. Each node token contains two sets of positional encoding. The first one represents the tree structure and is similar to the positional encoding from Theorem 4.1. The second one encodes the position in the reverse BFS ordering of the nodes. The role of this embedding is so that at each operation of the transformer, the last token that was generated will attend to the next token in the reverse BFS order. This way, all the sub-question in the same layer are solved one after the other, before moving on to the next layer. The construction of the transformer itself contains 2-layers of self-attention. The first layer is similar to a single layer of the construction from Theorem 4.1. Its role is to solve a single layer of sub-questions. The second layer make sure that the last generated token can only attend to the next token in the reverse BFS order, using the designated positional encodings.

Our work leaves open the question of whether a constant-depth transformer can solve CRQs by generating $o(n)$ CoT tokens. In Li et al. [2024] it was shown that transformers with constant depth that generate $O(\log(n))$ CoT tokens lie in $TC^0$ (see also Theorem 4 of Merrill and Sabharwal [2025]). By Lemma 4.2 we know that solving CRQs is $NC^1$-hard. Thus, under the assumption that $TC^0 \neq NC^1$, we know that transformers need more than logarithmically many CoT tokens to solve

CRQs. However, the gap between $\Omega(\log(n))$ and our solution with $O(n)$ CoT tokens is still open. We also note that by Amiri et al. [2025], in the finite precision case, a lower bound of $\Omega(n)$ CoT tokens for solving CRQs can be derived. Hence, in this case, our construction is asymptotically optimal. In the log-precision case (which is the main setting of this paper), the question of whether our construction is optimal remains open.

We now compare our result with the CoT solution from Feng et al. [2024] for arithmetic problems. Both solutions use a transformer with $O(1)$ layers and $O(\log(n))$ bit precision. However, their solution generated $O(n^2)^4$ CoT tokens, while ours generates only $O(n)$ tokens. This is a significant improvement, since for large sequences, a quadratic number of generated tokens can be infeasible to generate during inference time. The reason for this improvement is that in our solution each generated token solves exactly one sub-question, thus the number of tokens can be bounded by the number of sub-questions. In the solution of Feng et al. [2024], each time one sub-problem is solved (e.g. adding two numbers) the entire expression is rewritten. Thus, the number of tokens needed is $n + (n-1) + (n-2) + \cdots = O(n^2)$. However, we use a different input format that encodes the tree structure directly into the positional encodings, rather than processing the arithmetic expression from left to right.

# 7   Discussion and Future Work

In this paper, we introduce the framework of compositional reasoning questions. We describe the trade-offs inherent in solving CRQs with deep transformers, RNNs, and shallow transformers with CoT. Our results indicate that although transformers are highly parallelizable, they must be deep to be able to solve CRQs; in particular, the depth must scale with the depth of the CRQ tree. In contrast, RNNs and CoT can solve CRQs with constant depth, but their operation is not parallelizable. Transformers require quadratic computational cost, but RNNs can solve CRQs in nearly linear time. Finally, transformers with CoT solve CRQs using a logarithmic number of parameters, while the other models require the number of parameters to be linear in $n$ in the worst case.

Our results have a broader significance for the grand challenge of improving LLM reasoning. The three architectures we consider correspond to three schools of thought about how to make progress on this problem, each of which has received support from the theoretical literature:

1. Scale up transformers by, e.g., adding more layers, and greater capabilities will continue to emerge. Transformers with depth $\log(n)$ can do k-hop induction and graph algorithms [Merrill and Sabharwal, 2025, Sanford et al., 2024b,c].

2. Devise better architectures, such as state space RNNs or hybrid models, because attention is not all you need to learn and reason efficiently. RNNs can more easily do state tracking tasks like recognizing regular languages [Merrill et al., 2024].

3. Introduce test-time scaling via chain of thought, opening a new axis of the design space. This enables dynamic programming and simulating Turing machines [Feng et al., 2024, Merrill and Sabharwal, 2023b].

In previous work, the school that looks most promising depends on which model problem they chose to study. CRQs are just as fundamental as any of the other problems studied in the literature. However, when using CRQs as a theoretical yardstick of reasoning, we find that none of the schools is a clear winner and none is a clear loser. Thus, the theory of representational capacity does not give a single answer to the question of how to improve LLM reasoning. Perhaps some blend of the three schools or a fourth school will emerge that dominates the others, but for now the answer is equivocal and contingent, *even for a single task*.

There are several open questions remaining. First, is our solution using constant depth transformers that generates $n$ CoT tokens optimal, or is it possible to solve CRQs while generating fewer tokens? Second, our work only considers the expressiveness point of view, it would be interesting to understand the optimization process of transformers and other models when solving CRQs. Finally, it would be interesting to understand how general the solutions learned by transformers are. Do they generalize to other trees with different structures, sizes and different question regimes?

---

[4] In the arithmetic problems from Feng et al. [2024] we refer to $n$ as the number of symbols in the problems. i.e. numbers, parenthesis and arithmetic operations.

**Acknowledgments**   NA was supported by NSF award 2234660. We thank Will Merrill for helpful discussions related to this work and for drawing our attention to the Boolean formula evaluation problem.

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

# A Proofs from Section 4

## A.1 Proof of Theorem 4.1

The following lemma is useful for defining the positional encodings:

**Lemma A.1.** *For any $k \geq 2$ there exist $\mathbf{v}_1, \ldots, \mathbf{v}_k \in \{\pm 1\}^{4 \log(k)}$ such that $|\langle \mathbf{v}_i, \mathbf{v}_j \rangle| \leq 3 \log(k)$ for all $i \neq j$.*

*Proof.* We sample $\mathbf{v}_1, \ldots, \mathbf{v}_k \in \{\pm 1\}^{4 \log(k)}$ uniformly at random. Note that for any $i \neq j$ we have that $\mathbb{E}[\langle \mathbf{v}_i, \mathbf{v}_j \rangle] = 0$. By Hoeffding's inequality we have that:

$$\Pr\left(|\langle \mathbf{v}_i, \mathbf{v}_j \rangle| \geq 3 \log(k)\right) \leq 2 \exp(-4 \log(k)) . \tag{4}$$

Applying union bound on the above for all pairs $i \neq j$ we get that:

$$\Pr\left(\forall i \neq j, \ |\langle \mathbf{v}_i, \mathbf{v}_j \rangle| \leq 3 \log(k)\right) \geq 1 - 2 \exp(-4 \log(k)) \cdot k^2 \geq 1 - \frac{2}{k^2} .$$

In particular, for $k \geq 2$ this probability is non-zero, meaning that there exists such vectors $\mathbf{v}_1, \ldots, \mathbf{v}_k$. $\square$

*Theorem 4.1.* For any CRQ with up to $n$ nodes, there exist at most $|Y| \leq n$ possible queries. By Lemma A.1 there exist vectors $\mathbf{z}_1, \ldots, \mathbf{z}_n \in \{\pm 1\}^{4 \log(n)}$ such that $|\langle \mathbf{z}_i, \mathbf{z}_j \rangle| \leq 3 \log(n)$ for any $i \neq j$.

We will first describe how we embed each node as an input token to the transformer, and then describe the construction of the transformer itself. For each node $v \in V$ that is not a leaf, we set one of the vectors $\mathbf{z}_i$'s to be corresponded with this node, and denote it as $\mathbf{z}_v$. Recall that $\mathcal{P}(v)$ is the parent of node $v$. Note that although each node may have several children, it only has one parent.

Each node of the tree will be embedded as a vector in $\mathbb{R}^{2d + 8 \log(n) + 2}$. The input tokens corresponding to a leaf node $u$ and a non-leaf node $v$ are defined as:

$$\mathbf{t}_u := \begin{pmatrix} \mathbf{x}_u \\ \mathbf{0}_d \\ \mathbf{z}_{\mathcal{P}(u)} \\ \mathbf{0}_{4 \log(n)} \\ 0 \\ 0 \end{pmatrix}, \quad \mathbf{t}_v := \begin{pmatrix} \mathbf{x}_v \\ \mathbf{0}_d \\ \mathbf{z}_v \\ \mathbf{z}_{\mathcal{P}(v)} \\ \ell(v) \\ \mathbb{1}(\ell(v) = L - 1) \end{pmatrix}, \tag{5}$$

where $\ell(v)$ is the depth of node $v$ (the length of the shortest path from $v$ to the root).

The crux of the construction lies mainly in the way we choose the embedding: The embedding first contains the value of the leaves or query of the corresponding node. The embedding also contains a positional encoding (PE), where each leaf token contains its parent's PE, and each non-leaf token contains its own PE as well as its parent's. In addition, there is a number representing the depth of each node and a flag on whether a node is in the second to last layer (i.e it is the deepest intermediate node connected to a leaf).

We construct the weights of the transformer so it will work as follows: In the first layer, each token representing an intermediate node will attend either to itself, if its depth is higher than $L - 1$, or to its children, if its depth is exactly $L - 1$. This means that only the tokens of nodes in depth $L - 1$ will change while all other tokens of intermediate nodes will remain the same. Using the positional encodings, the tokens at depth $L - 1$ will attend only to their children, not to themselves, and after the self-attention layer (before applying the $V$ matrix) will be equal to the token representing their child with the highest correlation to their query. We now use the $V$ matrix, residual connection and MLP so that the tokens with depth $L - 1$ will have a similar embedding to the leaves tokens, and decrease the depth of all other intermediate tokens. We also make sure that the flag after the first layer will be equal to 1 for nodes of depth $L - 2$. We now apply a similar layer exactly $L$ times, so that in the final layer the output of the token representing the root will be equal to the solution to the CRQ.

We now turn to the formal construction of the transformer. The order of the input tokens is not important for the construction, except for the root token which will be the last token since its final

embedding will include the answer to the CRQ. Let $c \geq 1$ be some universal constant such that $\|\mathbf{x}\|^2 \leq c$ for every $\mathbf{x} \in \Gamma^d$, there exists such a constant since $\Gamma$ is finite. The matrices of all the layers of the transformer will be the same and equal to:

$$K = \begin{pmatrix} I_{2d+8\log(n)} & & \\ & 0 & \\ & & 1 \end{pmatrix},$$

$$Q = \begin{pmatrix} I_{2d} & & & & \\ & 6cI_{4\log(n)} & & & \\ & & cI_{4\log(n)} & & \\ & & & 0 & \\ & & & & -6c\log(n) \end{pmatrix},$$

$$V = \begin{pmatrix} 0_d & I_d & \\ I_d & 0_d & \\ & & 0_{8\log(n)+2} \end{pmatrix}.$$

Before defining the MLP, we will explain how the attention layer operates on the input tokens. Since the attention mechanism contains a hardmax head, the output of each token will depend only on the token to which it attends. We will first show the next two claims:

1. Each token non-leaf node $v$ with $(\mathbf{t}_v)_{-1} = 0$ will attend to itself.

2. Each token non-leaf $v$ with $(\mathbf{t}_v)_{-1} = 1$ will attend to $\mathbf{t}_u$ with $\arg\max_{\mathbf{t}_u, u \in \mathcal{C}(v)} \langle \mathbf{x}_u, \mathbf{x}_v \rangle$.

For the first claim, let $v$ be some node with $(\mathbf{t}_v)_{-1} = 0$. Then we have that:

$$\mathbf{t}_v^\top K^\top Q \mathbf{t}_v = \|\mathbf{x}_v\|^2 + 6c\|\mathbf{z}_v\|^2 + c\|\mathbf{z}_{\mathcal{P}(v)}\|^2 \geq 28c\log(n) \tag{6}$$

Now, let $u \neq v$ be some other node. If $u$ is a leaf then we have that:

$$\mathbf{t}_u^\top K^\top Q \mathbf{t}_v \leq |\langle \mathbf{x}_u, \mathbf{x}_v \rangle| + 6c|\langle \mathbf{z}_{\mathcal{P}(u)}, \mathbf{z}_v \rangle| + c|\langle \mathbf{0}, \mathbf{z}_{\mathcal{P}(v)} \rangle|$$
$$\leq c + 24c\log(n) + < 25c\log(n).$$

If $u$ is not a leaf then we have that:

$$\mathbf{t}_u^\top K^\top Q \mathbf{t}_v \leq |\langle \mathbf{x}_u, \mathbf{x}_v \rangle| + 6c|\langle \mathbf{z}_u, \mathbf{z}_v \rangle| + c|\langle \mathbf{z}_{\mathcal{P}(u)}, \mathbf{z}_{\mathcal{P}(v)} \rangle|$$
$$\leq c + 18c\log(n) + 4c\log(n) < 23c\log(n),$$

where we used that $|\langle \mathbf{z}_u, \mathbf{z}_v \rangle| \leq 3\log(n)$ for $u \neq v$. This shows that $\mathbf{t}_v$ can only attend to itself.

Next, let $v$ be a node with $(\mathbf{t}_v)_{-1} = 1$. Let $u$ be a node with $u \in \mathcal{C}(v)$ and let $w \neq v$ with $w \notin \mathcal{C}(v)$. We have that:

$$\mathbf{t}_v^\top K^\top Q \mathbf{t}_u = \langle \mathbf{x}_u, \mathbf{x}_v \rangle + 6c\|\mathbf{z}_v\|^2$$
$$\geq -c + 24c\log(n) \geq 23c\log(n)$$
$$\mathbf{t}_v^\top K^\top Q \mathbf{t}_w \leq |\langle \mathbf{x}_w, \mathbf{x}_v \rangle| + 6|\langle \mathbf{z}', \mathbf{z}_v \rangle| + |\langle \mathbf{z}_{\mathcal{P}(w)}, \mathbf{z}_{\mathcal{P}(v)} \rangle|$$
$$\leq c + 18c\log(n) + 4c\log(n) < 23c\log(n)$$
$$\mathbf{t}_v^\top K^\top Q \mathbf{t}_v = \|\mathbf{x}_v\|^2 + 6c\|\mathbf{z}_v\|^2 + c\|\mathbf{z}_{\mathcal{P}(v)}\|^2 - 6c\log(n)$$
$$\leq c + 24c\log(n) + 4c\log(n) - 6c\log(n) < 23c\log(n).$$

Here $\mathbf{z}'$ is some vector with $\mathbf{z}' \neq \mathbf{z}_v$ (n fact, $\mathbf{z}' = \mathbf{z}_w$ or $\mathbf{z}' = \mathbf{z}_{\mathcal{P}(w)}$, depending on whether $\mathbf{w}$ is a leaf). This shows that $\mathbf{t}_v$ can only attend to its children. It is also clear that $\mathbf{t}_v$ will attend to its child $u$ that maximized $\langle \mathbf{x}_u, \mathbf{x}_v \rangle$. This finishes the two claims above.

After applying the $V$ matrix and the residual connection from the previous layer, the output of the self-attention layer on token $\mathbf{t}_v$ is equal $\mathbf{t}_v + \mathbf{o}_v$, where $(\mathbf{o}_v)_{(d+1:2d)} = \mathbf{x}_v$ if $(\mathbf{t}_v)_{-1} = 0$ and $(\mathbf{o}_v)_{(d+1:2d)} = \arg\max_{\mathbf{x}_u, u \in \mathcal{C}(v)} \langle \mathbf{x}_u, \mathbf{x}_v \rangle$ if $(\mathbf{t}_v)_{-1} = 1$. In words, the entries of each token in place $d+1$ until $2d$ after the self-attention layer is the solution to the sub-question for the deepest non-leaves nodes of the tree, and for other nodes it is the queries themselves.

We now define an MLP that will reorganize the tokens before applying the next layer of self-attention. The role of the MLP will be to perform the following operations on an input token $\mathbf{t}_v$:

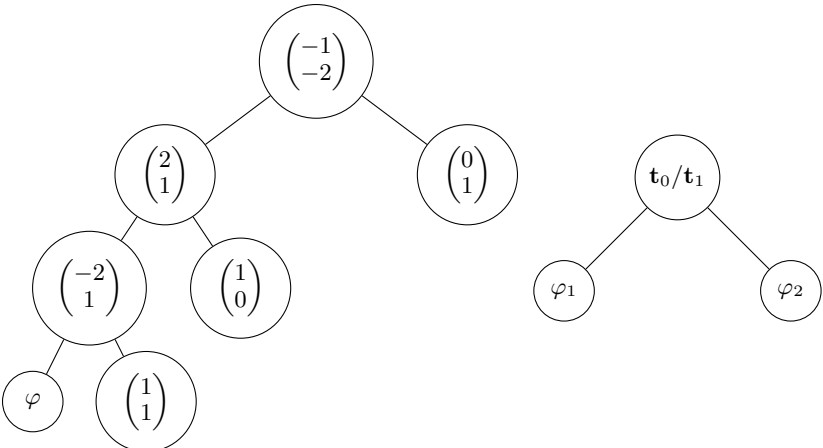

Figure 3: Left: Construction for $\neg\varphi$. Right: Construction for $\varphi_1 \wedge \varphi_2$ and $\varphi_1 \vee \varphi_2$

1. If the entries in places $2d + 4\log(n) + 1$ to $2d + 8\log(n)$ are zero, then $\mathbf{t}_v = \mathbf{0}$.

2. If the last entry of $\mathbf{t}_v$ is equal to 1, then replace the entries in places $2d+1$ until $2d+4\log(n)$ (which used to hold $\mathbf{z}_v$) with the entries in places $2d + 4\log(n) + 1$ until $2d + 8\log(n)$ (which used to hold $\mathbf{z}_{\mathcal{P}(v)}$). Then, replace the entries in places $2d + 4\log(n) + 1$ until $2d + 8\log(n)$ with zeroes. This makes it a leaf embedding.

3. Replace between the entries in places $1$ until $d$ and $d + 1$ until $2d$. Then replace the entries in places $d + 1$ until $2d$ with zeroes.

4. Add 1 to the entry in place $2d + 8\log(n) + 1$. Then, if this entry is equal to $L - 1$, replace the last entry with 1, if this entry is equal to $L$ replace the last two entries with zeroes.

The first operation zero out any token that corresponded to a leaf node, as they won't be necessary in later stages of the computation. The second operation turn tokens in the second to last layer to have the same embedding as leaf nodes, namely that they only contain the PE of their parent, and don't have a PE of their own. The third operation update the value of $\mathbf{x}_v$, this only changes for nodes that were in the second to last layer. For any other non-leaf node, since it attended to itself both vectors are equal. The last operation updates the depth of each node. Now, each node that was in the second to last layer will be embedded similarly to a leaf, while nodes in the third to last layer will have 1 in their last coordinate.

Each such operation can be implemented by a 2-layer MLP with width bounded by $O(d + \log(n))$. For example, to implement the first operation we use the function:

$$f(z) = \sigma(z + 1) - \sigma(z - 1/2) + \sigma(1 - z) - \sigma(1/2 - z) \,,$$

that can be implemented by the first layer of an MLP. We concatenate the output of this function on the $2d + 4\log(n) + 1$-th coordinate of the input, to the input itself. Then, we use the output layer to apply the function $\tilde{\mathbf{x}} \mapsto \mathbf{0}_k \cdot (\tilde{\mathbf{x}})_{k+1} + \tilde{\mathbf{x}} \cdot (1 - (\tilde{\mathbf{x}})_{k+1})$, where $\tilde{\mathbf{x}}$ is the input concatenated with the additional coordinate of the output of $f$, and $k = 2d + 8\log(n) + 2$. use similar constructions to implement the rest of the operations.

After applying the MLP we have that each token is either equal to: (1) $\mathbf{0}$ if it was previously a leaf. (2) An embedding of a leaf if it was previously in the second to last layer; or (3) An embedding of a non-leaf node if it wasn't a leaf or in the second to last layer. By applying the same construction $L$ times, we get that each node except the root is equal to $\mathbf{0}$, while the first $d$ coordinates of the root contain the answer to the CRQ, which is the final output of the transformer. This finishes the proof. $\qquad\square$

## A.2 Proof of Lemma 4.2

Our proof relies on a reduction to the Boolean Formula Evaluation Problem (BFEP). It was shown in Buss [1987], that this problem is $\mathsf{NC}^1$-complete, thus a reduction of BFEP to CRQ would finish the

proof. The BFEP is defined recursively over the alphabet: $\Sigma = \{1, 0, \vee, \wedge, \neg, (,)\}$ in the following way:

1. $0$ and $1$ are Boolean formula.
2. If $\varphi$ is a Boolean formula, then $(\neg\varphi)$ is a Boolean formula.
3. If $\varphi_1, \varphi_2$ are Boolean formulas, then $(\varphi_1 \vee \varphi_2)$ and $(\varphi_1 \wedge \varphi_2)$ are Boolean formulas.

The goal in BFEP, is given a Boolean formula to evaluate whether it is true or false (i.e. outputs $1$ or $0$). We will construct a translation function $f$ from Boolean formulas to CRQ, and specifically to a tree structure where each node is given a specific value. We will use vectors in $\mathbb{R}^2$, while in fact all the possible values of the vectors in the CRQ will be in $\{0, \pm 1, \pm 2\}$. We define the vectors $\mathbf{t}_1 = \begin{pmatrix} 1 \\ 0 \end{pmatrix}$ and $\mathbf{t}_0 = \begin{pmatrix} 0 \\ 1 \end{pmatrix}$. The translation function is defined recursively as follows:

1. For $f(0)$ and $f(1)$ we construct a node in the tree whose corresponding value is equal to $\mathbf{t}_0$ and $\mathbf{t}_1$ respectively.
2. Given two Boolean formulas $\varphi_1, \varphi_2$ we define $f((\varphi_1 \wedge \varphi_2))$ and $f((\varphi_1 \vee \varphi_2))$ as the tree in Figure 3 (right), where for $\wedge$ we use $\mathbf{t}_0$ for the root, and for $\vee$ we use $\mathbf{t}_1$.
3. Given a Boolean formulas $\varphi$ we define $f((\neg\varphi))$ as the tree in Figure 3 (left).

We will first show that this construction can be done using $\mathsf{TC}^0$ circuits, and then show its correctness. Given a Boolean formula of length $s$, it is clear that the construction above creates a tree with at most $O(|s|)$ nodes, where the values of each nodes is a 2-d vectors with entries in $\{0, \pm 1, \pm 2\}$. Thus, the total number of bits requires for the construction is also bounded by $O(|s|)$.

The translation works recursively over the logical operators $\neg, \wedge, \vee$, and to make the construction work we need to know in which order to translate them. Namely, the root should begin with the outmost logical operator, its children the second to outmost operators etc., until we reach either $0$ or $1$, which should be translated to leaves in the tree. We will show that this order can be determined using $\mathsf{TC}^0$ circuits. Given a Boolean formula $\varphi$ we define by $\varphi_i$ its character in place $i$. We define:

$$r_i = \sum_{j < i} \mathbb{1}\left[\varphi_j = `(` \right] - \sum_{j < i} \mathbb{1}\left[\varphi_j = `)` \right]. \tag{7}$$

It is easy to see that $r_i$ can be calculated using $\mathsf{TC}^0$ circuits. Now, the order in which the recursion works is in increasing order w.r.t $r_i$ over the logical operators. Namely, begin with the logical operator in place $i$ where $r_i = 0$, continue with $r_i = \pm 2$ an so forth.

For the correctness of the reduction, it is an easy calculation to see that given $t_i$ for $i \in \{0, 1\}$ the construction for $\neg\varphi$ outputs $t_{1-i}$ with the definition of a CRQ. Also, for $\wedge$ and $\vee$ the construction outputs correctly $t_i \wedge t_j$ and $t_i \vee t_j$ for $i, j \in \{0, 1\}$. Therfore, this construction forms a reduction from the BFEP to CRQ, and thus CRQ is $\mathsf{NC}^1$-hard.

## B   Memory-rank sort

In this section we formally define the memory-rank sort. It is presented in Algorithm 1. We next prove that performing memory-rank sort and calculting the memory-rank of all the nodes in a tree can be done efficiently:

**Lemma B.1.** *Let $T = (V, E)$ be a rooted binary tree, then calculating the memory-rank of each node and performing the memory-rank sort takes time $O(|V|)$.*

*Proof.* Calculating the memory rank of each node can be done using a reverse breadth-first search (BFS) of the tree, traversing from the leaves to the root. BFS can be done in time $O(|V|)$, and calculating the memory rank of each nodes requires only knowledge of the memory rank of its children. Thus, this can be done in a linear time by going over each node only once.

The time that Algorithm 1 runs is the number of iterations in the "while" loop. Note that each leaf in the tree is visited at most once, since if $v_{\text{cur}}$ is a leaf, it is inserted to the list and not visited again. For any non-leaf node, it is visited at most 3 times during the loop. Once for each one of its children, and

---

**Algorithm 1:** Memory-Rank Sort

---

**Input:** A Tree $T = (V, E)$ with root $v_r$ and memory rank calculated for each node.

$S = []$

$v_{\text{cur}} \leftarrow v_r$

**while** $|S| \neq |V|$ **do**

    **if** $\mathcal{C}(v_{cur}) = \varnothing$ ***or*** $\mathcal{C}(v_{cur}) \subseteq S$ **then**

        $S \leftarrow v_{\text{cur}}$

        $v_{\text{cur}} \leftarrow \mathcal{P}(v_{\text{cur}})$

        **Continue**

    $v_{\text{cur}} \leftarrow \underset{v \in \mathcal{C}(v_{\text{cur}}), v \notin S}{\arg\max} \ \mathbf{mr}(v)$

**Return:** $S$

---

again when it is inserted into the list. In total the running time of the sorting algorithm is bounded by $3|V|$. $\qquad\qquad\square$

## C   Proofs from Section 5

### C.1   Proof of Theorem 5.2

*Proof.* Our proof uses the following claim from communication complexity:

**Claim C.1** (Lower bound for set disjointness [Yao, 1979])**.** *Suppose Alice and Bob are given inputs $a, b \in \{0, 1\}^n$, respectively. Their goal is to calculate $\max_i a_i b_i$ by alternately sending $1$-bit messages to one another over a sequence of communication rounds. Any deterministic protocol for computing $\max_i a_i b_i$ requires at least $n$ rounds of communication.*

For simplicity of the proof, we use trees with $4n - 1$ nodes instead of $n$, this will only affect the constant in the $\Omega$ notation. We construct the following CRQ: The tree $T = (V, E)$ is a balanced binary tree with $4n - 1$ nodes, thus it has $2n$ leaves. All the values of the leaves are in $\{\pm 1\}$. The nodes in the second to last layer are all equal to $-1$, and the rest non-leaf nodes are equal to $1$. We number the leaves as $v_1, \ldots, v_{2n}$. Suppose we are given an instance of the set-disjointness problem, where Alice is given the leaves $v_{2i-1}$ and Bob the leaves $v_{2i}$ for $i \in [n]$. By the definition of CRQ, the answer to the second to last layer nodes with leaves $v_{2i-1}$ and $v_{2i}$ is $1$ if $v_{2i-1} = v_{2i} = 1$ and $-1$ otherwise. In addition the answer to any other node will be $1$ if one of its children is $1$ and $-1$ otherwise. In total, the answer to the tree is $1$ if there exists some $i \in [n]$ with $v_{2i-1} = v_{2i} = 1$ and $-1$ otherwise.

Consider the following ordering of the nodes for the RNN: The first $n$ nodes are $v_{2i-1}$ for $i \in [n]$, the next $n$ nodes are $v_{2i}$ for $i \in [n]$. The root is the last node in this ordering, and the rest of the nodes are given in some arbitrary order. Suppose there is an RNN known to both Alice and Bob that solves the CRQ problem above in the prescribed order of the nodes, and it has a hidden state $\mathbf{h} \in \mathbb{R}^m$ where each coordinate is represented by $p$ bits. We will define the following communication protocol to solve the set disjointness problem: Alice apply the RNN on her inputs. After each input $v_{2i-1}$ for $i \in [n]$ she passes the hidden state $\mathbf{h}_i$ to the next recurrence of the RNN. After $n$ such recurrences of the RNN, the hidden state $\mathbf{h}_n$ is passed to Bob, he inputs his inputs $v_{2j}$ for $j \in [n]$ in the prescribed order, each time passing a hidden state. He continue to run the RNN on the rest of the nodes, and output the answer to the CRQ. The number of bits transferred between Alice and Bob is $m \cdot p$, since the only communication between them is transferring $\mathbf{h}_n$ and Alice finishes processing her inputs. By Claim C.1 we have that $mp = \Omega(n)$, which finishes the proof.

$\qquad\qquad\square$

### C.2   Proof of Theorem 5.4

We first need the following two lemmas:

**Lemma C.2.** *For a binary tree $T$ with $n$ nodes, we have that $\mathbf{mr}(T) \leq \log(n)$.*

*Proof.* Let $r_k$ be the smallest number of descendants for a node with memory-rank $k$. It is easy to see that $r_1 = 2$, which happens when the node has only two children that are leaves. Let $v$ be a node with $\mathbf{mr}(v) = k$, and let $u_1, u_2$ be its children. It can be seen that $r_k \geq 2r_{k-1}$. This is because, if $\mathbf{mr}(u_1) \neq \mathbf{mr}(u_2)$, then one of them must be equal to $k$, assume w.l.o.g it is $u_1$. This means that either $u_1$ or one of its descendants have two children with a memory-rank of $k-1$. In particular, the number of descendants of $v$ is larger than $2r_{k-1}$. If $\mathbf{mr}(u_1) = \mathbf{mr}(u_2)$, then they are both equal to $k-1$, in which case the same conclusion follows. Applying the recurrence formula, we get that $r_k \geq 2^k$.

Let $k$ be the memory rank of a tree with $n$ nodes. We have $2^k \leq r_k \leq n$. Thus, $k \leq \lfloor \log(n) \rfloor$. Hence, for any tree with $n$ nodes, the largest possible memory-rank of the tree is bounded by $\log(n)$.

$\square$

**Lemma C.3.** *Let $R, \epsilon > 0$ and $d \in \mathbb{N}$. There exists a 2-layer neural network $\mathcal{N} : \mathbb{R}^{2d} \to \mathbb{R}$ with width $O\left(\frac{d^2 R}{\epsilon}\right)$ such that $\max_{\mathbf{x}, \mathbf{y} \in [-R, R]^d} |\mathcal{N}(\mathbf{x}, \mathbf{y}) - \langle \mathbf{x}, \mathbf{y} \rangle| \leq \epsilon$.*

*Proof.* We can use Lemma 6 from Daniely [2017] to find a 2-layer network $N : \mathbb{R}^2 \to \mathbb{R}$ with width $O\left(\frac{R}{\epsilon}\right)$ such that $\max_{x, y \in [-R, R]^d} |N(x, y) - x \cdot y| \leq \epsilon$. Summing $d$ such networks over the coordinates of $\mathbf{x}$ and $\mathbf{y}$, and replacing $\epsilon$ with $\epsilon' = \frac{\epsilon}{d}$ proves the lemma. $\square$

*Theorem 5.4.* The main crux of the proof is to simulate a stack using a ReLU network with constant depth. The maximal size of the stack for a tree $T = (V, E)$ will be $O(d \cdot \mathbf{mr}(T))$. The RNN will execute the following pseudo-code:

1. Given an input node $v$, check the last node $u$ in the stack:

    (a) If $u$ has the same depth or a larger depth than $v$, then add $v$ to the stack.
    (b) If $u$ has a depth that is lower by exactly 1 than $v$, then pop out the last two nodes from the stack $u$ and $w$. If $\langle u, v \rangle \geq \langle w, v \rangle$ then insert $u$ into the stack and raise its depth by 1. Otherwise, inset $w$ and raise its depth by 1.

2. The stack is the hidden state that is transferred to the next iteration of the RNN.

We now turn to the formal construction. Each node $v \in V$ in the tree will be embedded as a vector $\begin{pmatrix} \mathbf{x}_v \\ \ell(v) \end{pmatrix} \in \mathbb{R}^{d+1}$, where $\ell(v)$ is the depth of $v$. The hidden state of the RNN at the beginning will be $\mathbf{h} = \mathbf{0}_{d(\log(n)+1)}$. The input to the RNN will be the current input vector, concatenated to the hidden state. Throughout the proof we refer to $\tilde{\mathbf{x}}$ the vector that is inputted to the RNN, namely the concatenation of the embedding of each node with the hidden state. We call the part of $\tilde{\mathbf{x}}$ that contains the hidden state $\mathbf{h}$ the stack, and each consecutive $d+1$ coordinates in it as a node that is was inputted to the stack. The reason is that this part of the input will simulate a stack where the RNN can only access its first and second nodes, namely the first $2d+2$ coordinates. The RNN will add at most 3 coordinates to $\tilde{\mathbf{x}}$, which will simulate flags with values that are either 0 or 1. The RNN contains the following layers:

1. **Layer 1:** Check whether the depth of the input node $v$ is the same or larger than the depth of the last node in the stack. If so, turn on a specific flag for this event.

2. **Layer 2:** If the flag is turned on, insert $v$ into the stack by moving all the elements $\log(n)+1$ entries forward and putting $v$ at the top of the stack.

3. **Layer 3:** If the flag is turned off, extract the last two nodes from the stack $u_1$ and $u_2$, and calculate $\langle \mathbf{x}_{u_1}, \mathbf{x}_v \rangle$, $\langle \mathbf{x}_{u_2}, \mathbf{x}_v \rangle$.

4. **Layers 4 & 5:** If the flag is turned off, insert $u_1$ or $u_2$ to the stack, whichever has the higher inner product with $v$. Also, increase the depth of this node by 1.

Note that the input to the RNN is of dimension $d(\log(n)+1)+d+1$, which is a concatenation of the embedding of the current input node and the hidden state. We will also need to following parameters:

Let $c > 0$ be such that $c > |a|$ for every $a \in \Gamma$, namely, it is larger than all possible entries of the $\mathbf{x}_v$'s, and that $c > L$, the depth of the tree. We have that $c = O(n)$ which is the largest possible depth of the tree. Also, let $\epsilon > 0$ such that $\min_{a,b \in \Gamma} |a - b| < \epsilon$. We think about $\epsilon$ as a constant independent of $n$ and $d$, as it only depends on the possible values in $\Gamma$. Each layer is constructed in the following way:

**First layer**: We denote the input vector to the RNN is $\tilde{\mathbf{x}}$. The first layer add a coordinate to the input. All the input coordinates are copied, while the last is constructed as: $\sigma((\tilde{\mathbf{x}})_{d+1} - (\tilde{\mathbf{x}})_{2d+1} + 1) - \sigma((\tilde{\mathbf{x}})_{d+1} - (\tilde{\mathbf{x}})_{2d+1})$. Note that this added last coordinate is equal to 1 if $(\tilde{\mathbf{x}})_{d+1} - (\tilde{\mathbf{x}})_{2d+1} \geq 0$, meaning that the current node has an equal or larger depth that the last node in the stack, and 0 if $(\tilde{\mathbf{x}})_{d+1} - (\tilde{\mathbf{x}})_{2d+1} \leq -1$. The copying of the input can be done by a ReLU network, since $\sigma(z) - \sigma(-z) = z$ for every $z \in \mathbb{R}$. Thus, adding two identity matrices to the weights of the MLP with different signs, and adding their outputs together will copy the inputs.

**Second layer:** Let $A = \begin{pmatrix} I_{d+1} & \mathbf{0} \\ I_{d(\log(n)+1)} & \mathbf{0} \end{pmatrix}$ be a $d(\log(n) + 1) + d + 1$ square matrix. The second layer will perform the following operation:

$$\mathbf{x} \mapsto \sigma\left(I\mathbf{x}_{(1:-2)} - 2cI\mathbf{x}_{(-1)} + c\mathbf{1}\right) + \sigma\left(A\mathbf{x}_{(1:-2)} - 2cI(-\mathbf{x}_{(-1)} + 1) + c\mathbf{1}\right) - c\mathbf{1} .$$

This operation can be implemented by a 2-layer network, where the constant $\mathbf{1}$ vectors are added using the bias terms. This operation will apply the identity matrix to the inputs if the flag (defined in the previous layer) is 0, and apply the $A$ matrix if it is 1. The $A$ matrix will copy the first $d + 1$ entries (the current input node), and will move also add it to the stack, while also moving all the other entries $d + 1$ coordinates further down in the stack. The added $c\mathbf{1}$ factor is so that if there are negative values in the entries of the inputs, they will not be removed by the ReLU, this factor is removed at the end to keep the original value. We will later on prove that the stack will not get overflown, meaning that we don't delete the last $d + 1$ entries of it.

**Third layer:** This layer will be used to approximate inner products. Assume that $v$ is a non-leaf node and let $u_1$ and $u_2$ be its two children. If it is a leaf node, this layer will not effect its input. The layer will add two coordinates to its input vector, corresponding to an approximation of $\langle \mathbf{x}_{u_1}, \mathbf{x} \rangle$ and $\langle \mathbf{x}_{u_2}, \mathbf{x} \rangle$.

By Lemma C.3 there exists a 2-layer neural network $\mathcal{N} : \mathbb{R}^{2d} \to \mathbb{R}$ with width $O(d^2)$ such that $\max_{\mathbf{x},\mathbf{y} \in \Gamma^d} |\mathcal{N}(\mathbf{x}, \mathbf{y}) - \langle \mathbf{x}, \mathbf{y} \rangle| \leq \frac{\epsilon}{4}$. We will stack two such networks that approximate up to an error $\frac{\epsilon}{4}$ the inner products $\langle \mathbf{x}_{u_1}, \mathbf{x}_v \rangle, \langle \mathbf{x}_{u_2}, \mathbf{x}_v \rangle$. If $\mathbf{x}_{u_1} \neq \mathbf{x}_{u_2}$, then by the assumption on $\epsilon$ we have that $|\langle \mathbf{x}_{u_1}, \mathbf{x}_v \rangle - \langle \mathbf{x}_{u_2}, \mathbf{x}_v \rangle| \geq \epsilon$. By the construction of $\mathcal{N}$, we also have that:

$$|\langle \mathbf{x}_{u_1}, \mathbf{x}_v \rangle - \langle \mathbf{x}_{u_2}, \mathbf{x}_v \rangle| \leq |\mathcal{N}(\mathbf{x}_{u_1}, \mathbf{x}_v) - \mathcal{N}(\mathbf{x}_{u_2}, \mathbf{x}_v)| + \frac{\epsilon}{2} .$$

This means that $|\mathcal{N}(\mathbf{x}_{u_1}, \mathbf{x}_v) - \mathcal{N}(\mathbf{x}_{u_2}, \mathbf{x}_v)| \geq \frac{\epsilon}{2}$. We use a similar construction to the previous layer to apply this calculation and add $\mathcal{N}(\mathbf{x}_{u_1}, \mathbf{x}_v)$ and $\mathcal{N}(\mathbf{x}_{u_2}, \mathbf{x}_v)$ only if the flag in the last coordinate is turned off. Otherwise, if the flag is turned on, we just copy the inputs.

**Fourth and Fifth layers:** In the last layer, we add an additional flag on whether $\langle \mathbf{x}_{u_1}, \mathbf{x}_v \rangle > \langle \mathbf{x}_{u_2}, \mathbf{x}_v \rangle$ and 0 otherwise. This can be calculated by using the following function:

$$\mathbf{x} \mapsto \frac{2}{\epsilon} \left( \sigma((\mathbf{x})_{-2} - (\mathbf{x})_{-2}) - \sigma((\mathbf{x})_{-2} - (\mathbf{x})_{-2} - \frac{\epsilon}{2}) \right) .$$

By the argument from the previous layer, this function will provide the additional flag. All the other coordinates are copied as is. We use another layer to pop out the first $2d + 2$ coordinates from the stack, meaning that all the entries in the stack are moved upward by $d + 1$ coordinates, and the first $d + 1$ are changed to either $\begin{pmatrix} \mathbf{x}_{u_1} \\ \ell(u_1) - 1 \end{pmatrix}$ or $\begin{pmatrix} \mathbf{x}_{u_2} \\ \ell(u_2) - 1 \end{pmatrix}$, depending on whether the new flag is 1 or 0. Again, we apply this operation only if the flag from the first layer is turned off, by using a similar construction to the second layer. The rest of the coordinates are copied, and the coordinates of the flags that were used throughout the computation are removed. The output will be the hidden state to the next recurrence with the dimension as the hidden state that was inputted.

**Proof of correctness:** First, it is clear from the construction that applying the RNN on a single input node will perform the pseudo-code described in the beginning of the proof. We will show the the construction outputs the correct answer to the CRQ. When a node is inputted to the RNN, according to the ordering of the memory-rank sort, there are three options:

1. Its depth is equal to the the depth of the first node in the stack. This can only happen to leaf nodes, in which case, according to the memory-rank sorting, the node in the stack must be a sibling of the inputted node.

2. Its depth is larger than the depth of the first node in the stack. In this case, the first node in the stack will be pushed down to the second place after applying the RNN.

3. Its depth is smaller by exactly 1 than the last node in the stack. In which case, it must be its parent, and the second node in the stack is its sibling. After applying the RNN, both nodes are popped out, and the node that correctly answer the query corresponding to $v$ is inputted back to the stack, but with an updated depth. In this case, all the descendants of the inputted node have already being processed by the RNN, and the next inputted node will necessarily be its parent or sibling.

By induction over the nodes of the tree, and using these three options we get that after applying the RNN to the root (which is the last inputted node), the only node left in the stack is the answer to the question given by the root, which is the answer to the CRQ.

We are left with showing that the stack does not overflow. We will show that the maximal number of nodes in the stack is bounded by $\mathbf{mr}(T) + 1$. Since the number of coordinates in the representation of each node is $d + 1$, using Lemma C.2 finishes the proof. Let $v$ be some node with $\mathbf{mr}(v) = k$. We will first show by induction on $k$ that at any point in time, before processing $v$, the maximal number of nodes in the stacks that are also descendants of $v$ is bounded by $k$. If $k = 0$ then $v$ is a leaf a doesn't have any children. For the induction step, assume $k > 0$, and let $u_1, u_2$ be the two children of $v$. If $\mathbf{mr}(u_1) = \mathbf{mr}(u_2)$, then by the definition of memory rank, they are both equal to $k - 1$. Assume w.l.o.g that $u_1$ is processed before $u_2$. Note that in this case, $u_2$ is processed only after all the descendants of $u_1$ are processed. Then, by the induction step, the maximal number of nodes in the stacks that are also descendants of $u_2$ is bounded by $k - 1$. Hence, the maximal number of nodes in the stack that are descendants of $v$ is $k$, which includes $u_1$. If $\mathbf{mr}(u_1) \neq \mathbf{mr}(u_2)$, assume w.l.o.g that $\mathbf{mr}(u_1) > \mathbf{mr}(u_2)$, then $u_1$ is processed before $u_2$ (resp. $u_2$ before $u_1$). In this case, by the induction case, again the number of nodes in the stack the are descendants of $v$, are either descendants of $u_1$, in which case no nodes that are descendants of $u_2$ have being processed, or descendants of $u_2$, in which case $u_1$ is in the stack, as well as at most $k - 1$ descendants of $u_2$. This finished the induction.

Now, we will show that the number of nodes that are not descendants of $v$ and are in the stack is bounded by $1 + \mathbf{mr}(T) - k$. This will be by induction over $\mathcal{P}^{(i)}(v)$, namely the ancestors of $v$. We assume that $\mathbf{mr}(k) = v$. If $\mathbf{mr}(\mathcal{P}(v)) = k$, then $v$ is processed before its sibling, which by the induction we showed before, the number of nodes in the stack that are descendant of $\mathcal{P}^{(i)}(v)$ is bounded by $k$. If $\mathbf{mr}(\mathcal{P}(v)) = k + 1$, then the sibling of $v$ either have a memory-rank of $k$ or larger than $k$. In both cases it can be processed before $v$ (as well as all its descendants), hence the number of nodes in the stack that are descendants of $\mathcal{P}(v)$ is bounded by $k + 1$. Using the same induction argument as before, applied for the sibling of $\mathcal{P}^{(i)}(v)$, we get that the number of nodes the are descendants of $v$ that are in the stack is bounded by $\mathbf{mr}(\mathcal{P}^{(i+1)}(v)) + 1$.

Combining the two inductive arguments above, we get that when $v$ is processed into the RNN, the number of nodes in the stack is bounded by $\mathbf{mr}(T) + 1$. This finishes the correctness proof.

By induction on $\mathbf{mr}(v)$. If $\mathbf{mr}(v) = 0$, then it is a leaf. By Algorithm 1, since $v$ doesn't have children, the only node that is processed before it can be its sibling. When its sibling was processed necessarily all its children were already processed, hence there is at most 1 node in the stack. Assume that if $\mathbf{mr}(u) = k$, then there are at most $k + 1$ nodes in the stack, and let $v$ with $\mathbf{mr}(v) = k + 1$. Denote by $u_1$ and $u_2$ the children of $v$, and assume w.l.o.g that $\mathbf{mr}(u_1) \geq \mathbf{mr}(u_2)$. If $\mathbf{mr}(u_1) > \mathbf{mr}(u_2)$, then $\mathbf{mr}(v) = \mathbf{mr}(u_1)$. This means that $u_1$ was processed before $u_2$. Hence, $u_1$ is in the stack, and $u_2$ was processed right before $v$, which by induction shows that there are at most $k + 1$ nodes in the stack. If $\mathbf{mr}(u_1) = \mathbf{mr}(u_2)$, the by the definition of memory-rank, it necessarily happen that they are both equal to $k$. Assume w.l.o.g that $u_1$ is processed before $u_2$ (the order of them being processed is chosen arbitrarily in this case). Hence, when processing $u_2$ there are at most $k$ nodes. □

## C.3 Proof of Theorem 5.5

We begin with proving the theorem for balanced binary trees and then generalize to other trees. Let $T$ be a rooted balanced binary tree of size $n$ with a given ordering of the nodes denoted as: $v_1, \ldots, v_n$. Let $k := \log_2(n+1)$. We define the vectors $\mathbf{t}_0 = \begin{pmatrix} 0 \\ 1 \end{pmatrix}$, $\mathbf{t}_1 = \begin{pmatrix} 1 \\ 0 \end{pmatrix}$, $\mathbf{t}_{\texttt{null}} = \begin{pmatrix} 0 \\ 0 \end{pmatrix}$. We now define a set of CRQs given by $\mathbf{x}_{v_i} \in \{\mathbf{t}_0, \mathbf{t}_1, \mathbf{t}_{\texttt{null}}\}$ for all $i \in [n]$. We will show that given an RNN that solves all CRQs with this tree and this ordering of the nodes must have a hidden state containing $\Omega(\log(n))$ bits.

As in Lemma 4.2, an intuitive way to look at these CRQs is as Boolean formulas. For leaf nodes $\mathbf{t}_0$ represents 0 and $\mathbf{t}_1$ represents 1. For non-leaf nodes, $\mathbf{t}_0$ represents $\wedge$ and $\mathbf{t}_1$ represents $\vee$. The vector $\mathbf{t}_{\texttt{null}}$ is used by a hidden node to pass values up the tree without changing them, as described below.

Let $u_1$ be the last leaf node in the prescribed ordering. Denote its ancestors as $u_2, \ldots, u_k$, where $u_k$ is the root. Also, for each $u_i$ with $i \neq 1$ denote its other child as $w_{i-1}$ (i.e. the sibling of $u_{i-1}$). Suppose that the current input to the RNN is $u_1$. We will show that the RNN needs at least $k$ bits in its memory at this point in the calculation to solve the CRQ.

We consider the set of CRQs defined by the following rules:

1. Each of the nodes $w_1$ and $u_1, \ldots, u_k$ is equal to either $\mathbf{t}_0$ or $\mathbf{t}_1$.

2. For each node $w_2, \ldots, w_{k-1}$, all its leaf descendants are identical to one another. Either they are all $\mathbf{t}_0$, or they are all $\mathbf{t}_1$. Its non-leaf descendants, including itself, are all equal to $\mathbf{t}_{\texttt{null}}$.

To solve the sub-question represented by $u_2$, the RNN must use 1-bit of memory to remember $w_1$, since by assumption, $w_1$ precedes $u_1$ in the sequence. Similarly, to solve the sub-question represented by $u_2$, the RNN must know the answer to $w_2$. Because of rule 2, each subquestion in the subtree rooted at $w_2$ results in a tie. Because all the leaves in this subtree are identical, the answer to the subquestion rooted at $w_2$ equals any of its descendant leaves. These leaves all precede $u_1$ in the sequence, so we must use 1-bit of memory to store their value. For each node $u_2, \ldots, u_k$, the RNN must store at least 1 bit of memory at the point when it processes $u_1$. In total, this amounts to $\log(n) - 1$ bits of memory that needs to be transferred through the hidden state of the RNN.

Now, let $T$ be some rooted tree of size $n$, which is not necessarily a balanced binary tree. Denote by $\ell := \mathbf{mr}(T)$, and recall this means that the root has a memory-rank of $\ell$. We will find a subset of $T$ that forms a balanced binary tree whose depth equals $\ell$. We will construct all the other nodes to simply copy their inputs up the tree, and appeal to the case of a balanced binary tree proved above.

We first show how to find such a subset. If the root has two children with a memory rank of $\ell - 1$, denote the root by $u_1$. Otherwise, exactly one of its children has a memory-rank of $\ell$; search recursively through *its* children until a node with two children of rank $\ell - 1$ is found and label it $u_1$. We now apply the above procedure recursively on the two children of $u_1$. This yields two nodes, $u_2$ and $u_3$, each of which has two children with a memory rank of $\ell - 2$. We apply this procedure recursively until we defined nodes $u_1, \ldots, u_{2^\ell}$, where each node is either a leaf or has two children with the same memory-rank as one another. By the definition of memory rank, this process will generate a subset that forms a (possibly non-consecutive) complete binary tree.

We now construct a family of CRQs on the original tree whose answer always equals that of the complete binary subset $\{u_1, \ldots, u_{2^\ell}\}$. Nodes in the subset are labeled with $\mathbf{t}_0, \mathbf{t}_1, \mathbf{t}_{\texttt{null}}$, as above, except we append an extra 1 to each of these vectors:

$$\mathbf{t}_0 = \begin{pmatrix} 0 \\ 1 \\ 1 \end{pmatrix}, \mathbf{t}_1 = \begin{pmatrix} 1 \\ 0 \\ 1 \end{pmatrix}, \mathbf{t}_{\texttt{null}} = \begin{pmatrix} 0 \\ 0 \\ 1 \end{pmatrix}, \tag{8}$$

Each node of the original tree that is not in the subset is labeled with one of the following two vectors: $\mathbf{t}_+ = \begin{pmatrix} 0 \\ 0 \\ 1 \end{pmatrix}$ or $\mathbf{t}_- = \begin{pmatrix} 0 \\ 0 \\ -1 \end{pmatrix}$. Specifically, leaf nodes that are not in the subset are labeled with $\mathbf{t}_-$, and non-leaf nodes are labeled with $\mathbf{t}_+$. To see that the answer to this CRQ equals that of the CRQ corresponding to the balanced binary subset, it suffices to show that the vectors $\mathbf{t}_+$ and $\mathbf{t}_-$ are never

chosen over $\mathbf{t}_0$ and $\mathbf{t}_1$ at any subquestion. First, we show that $\mathbf{t}_-$ is never chosen over $\mathbf{t}_0$ and $\mathbf{t}_1$. This can be seen be enumerating all the possible values that the parent (non-leaf) node can take on $(\mathbf{t}_0, \mathbf{t}_1, \mathbf{t}_{\texttt{null}}, \mathbf{t}_+)$ and seeing that each prefers $\mathbf{t}_0$ and $\mathbf{t}_1$ to $\mathbf{t}_-$. Second, $\mathbf{t}_+$ is never even compared to $\mathbf{t}_0$ and $\mathbf{t}_1$, because no leaves are labeled with $\mathbf{t}_+$.

We now define a CRQ over $u_1, \ldots, u_{2\ell}$ the same way as in the balanced binary tree case using $\mathbf{t}_0, \mathbf{t}_1, \mathbf{t}_{\texttt{null}}$. By construction, the depth of this synthetic tree is $\ell = \mathbf{mr}(T)$, so the RNN needs memory of size $\mathbf{mr}(T)$.

Finally, if an RNN solves all CRQs on trees of size $n$, it also solves them on balanced binary trees of size $n$. Thus, the number of required bits in the hidden state of the RNN is $\Omega(\log(n))$.

# D    Extension of the RNN solution to non-binary trees

## D.1    Converting non-binary to binary trees

Suppose we have a CRQ over a tree $T$ with $n$ nodes and max degree $k$. We can transform it to an equivalent CRQ over a binary tree $T'$ with $2n$ nodes and max degree $2$. By equivalent we mean that the answer to the two CRQs are the same.

The construction is as follows: Let $v$ be some node with degree $k$ and denote its leaves by $u_1, \ldots, u_k$. Suppose that $k = 2^\ell$ for some integer $\ell$. We create a balanced binary tree with at most $2k$ nodes and depth $\ell$. The leaves will have vectors $\mathbf{x}_{u_1}, \ldots, \mathbf{x}_{u_k}$ and all the non-leaf nodes have a vector equal to $\mathbf{x}_v$. It is easy to see that the answer to the sub-question of the root of this tree is the same as the answer to the sub-question of the node $v$ from the original tree. The number of nodes is increased by a factor of at most $2$. If $k$ is not a power of $2$, we can duplicate the last leaf node until the number of leaves is a power of $2$ and use the same construction.

Doing this for every node increases the size of the tree by a factor of at most $2$, and the resulting CRQ will have the same answer as the original CRQ.

## D.2    Memory-rank for non-binary trees

In this sub-section we explain how to extend some of the results from Section 5 to non-binary trees. We first define memory-rank for general trees:

**Definition D.1** (Memory Rank for non-binary trees). *Let $T = (V, E)$ be a rooted tree. The **memory rank** of a node $v \in V$ is defined recursively as: $\mathbf{mr}(v) = 0$ if $v$ is a leaf. If $v$ is not a leaf, suppose it has $k$ children denoted by $u_1, \ldots, u_k$ where $\mathbf{mr}(u_1) \geq \cdots \geq \mathbf{mr}(u_k)$, then:*

$$\mathbf{mr}(v) = \max\left(\mathbf{mr}(u_1), \mathbf{mr}(u_2) + 1, \ldots, \mathbf{mr}(u_k) + k - 1\right)$$

It is straightforward to see that this definition generalizes Definition 5.3. Also, Algorithm 1 readily generalizes to Definition D.1, since it has no dependence on the degree of each node.

Extending Theorem 5.4 to general trees with maximal degree of $k$ can be done in a straightforward way by having an RNN with hidden dimension of $O(dk \log(n))$ and $O(k)$ layers. The idea is to add for each node an additional coordinate representing its degree. Now, in the proof of Theorem 5.4, when the last two nodes are extracted from the stack (layer 3 in the construction), we split this into $k$ layers. Each layer checks the degree of the parent node from $2$ until $k$ and for degree $i$ extracts the last $i$ nodes from the stack. This can be implemented by a ReLU network of depth $O(k)$. Finally, the node with the highest inner product to the parent is inserted back to the stack with an updated depth (similarly to layers $4$ and $5$ from the proof of Theorem 5.4).

We believe it is also possible to extend this construction to having depth $O(1)$ and hidden dimension $O(d \log(n))$ for general trees by changing the sorting algorithm. This can be done by splitting the argmax operation over $k$ inputs into $k - 1$ argmax operations, each over $2$ inputs, where these operations are nested. We leave this construction for future works.

# E   Proofs from Section 6

The main idea of the construction is to iteratively use the self-attention layer constructed in Theorem 4.1, while wrapping it with a construction that allows outputting CoT tokens in the right order.

By Lemma A.1 there exist vectors $\mathbf{z}_1, \ldots, \mathbf{z}_n \in \{\pm 1\}^{4\log(n)}$ such that $|\langle \mathbf{z}_i, \mathbf{z}_j \rangle| \leq 3\log(n)$ for any $i \neq j$. We will use these vectors twice, and for two different use cases. The first one is similar to the construction of Theorem 4.1. Namely, for each node $v \in V$ that is not a leaf, we set one of the vectors $\mathbf{z}_i$ to be corresponded with this node and denote it as $\mathbf{z}_v$.

We also assume that the tree has a reverse BFS ordering. This means that we do a standard BFS ordering of the tree, and order of node $i$ in the reverse order is $n - i + 1$. For a node $v$, let $I(v)$ be be its place in the reverse BFS ordering. We also correspond one of the vectors defined above to each place in this order we denote this vector as $\mathbf{w}_{I(v)}$. For the root $v_r$ we have $I(v_r) = n$, and define $\mathbf{w}_{I(v_r)+1} = \mathbf{w}_1$. We use a different notation for the BFS ordering to not confuse them with the other set of positional encoding, although this is the same set of vectors. It is also possible to find another set of vectors with the same property, although it won't matter for the construction. The embedding for a leaf node $u$ and a non-leaf node $v$ are defined as:

$$
\mathbf{t}_u := \begin{pmatrix} \mathbf{x}_u \\ \mathbf{0}_d \\ \mathbf{z}_{\mathcal{P}(u)} \\ \mathbf{0}_{4\log(n)} \\ \mathbf{w}_{I(u)} \\ \mathbf{w}_{I(u)+1} \\ 0 \end{pmatrix} , \quad \mathbf{t}_v := \begin{pmatrix} \mathbf{x}_v \\ \mathbf{0}_d \\ \mathbf{z}_v \\ \mathbf{z}_{\mathcal{P}(v)} \\ \mathbf{w}_{I(v)} \\ \mathbf{w}_{I(v)+1} \\ 1 \end{pmatrix} \in \mathbb{R}^{2d+16\log(n)+1} . \tag{9}
$$

As in Theorem 4.1, all the coordinates of the positional encoding are in $\{\pm 1\}$, and the matrices that operate over these coordinates will have all their entries in $\{-1, 0, 1\}$. We will first give an intuitive explanation of the embedding and how the transformer will work. The embedding first contains the vector $\mathbf{x}$ corresponding to the node. It also contains its own positional encoding $\mathbf{z}_v$ if its a non-leaf, and the positional encoding of its parent. Next, it contains the vector representing its place in the reverse BFS ordering, as well as the subsequent vector in the order. The last coordinate is a flag for whether a node is a leaf.

The transformer will contain 2 self-attention layers, and an MLP with $O(1)$ layers after each self-attention layer. The first self-attention layer and the MLP afterwards is similar to the construction in Theorem 4.1. Their goal is to solve the sub-questions in the CRQ. This layer will use the $\mathbf{z}$'s positional encoding vectors. The second self-attention layer will make sure that the last CoT token that was created will attend to the next token corresponding to the node after it in the reverse BFS ordering. This way, each CoT token will provide an answer to one sub-question of the CRQ, and using the reverse BFS order we make sure that any subsequent questions will have the questions of their children already solved.

We now turn to the formal construction: Let $c \geq 1$ be some constant such that $\|\mathbf{x}\|^2 \leq c$ for every $\mathbf{x} \in \Gamma^d$. The matrices for the first self-attention layer are equal to:

$$
K = \begin{pmatrix} I_d & & & & \\ & \mathbf{0}_{d \times d} & & & \\ & & \sqrt{c} \cdot I_{4\log(n)} & & \\ & & & \mathbf{0}_{12\log(n) \times 12\log(n)} & \\ & & & & \sqrt{3c} \end{pmatrix}
$$

$$
Q = \begin{pmatrix} I_d & & & & \\ & \mathbf{0}_{d \times d} & & & \\ & & \sqrt{c} \cdot I_{4\log(n)} & & \\ & & & \mathbf{0}_{12\log(n) \times 12\log(n)} & \\ & & & & -\sqrt{3c} \end{pmatrix} ,
$$

$$
V = \begin{pmatrix} \mathbf{0}_{d \times d} & I_d & \\ I_d & \mathbf{0}_{d \times d} & \\ & & \mathbf{0}_{16\log(n) \times 16\log(n)+1} \end{pmatrix} .
$$

We will first show to which each token can attend to. Let $v$ be some non-leaf node, and $u$ a child of $v$ which is a leaf, and $w$ some other node. We will show that $v$ can only attend to $u$:

$$\mathbf{t}_v^\top K^\top Q \mathbf{t}_u = \langle \mathbf{x}_v, \mathbf{x}_u \rangle + c \left\| \mathbf{z}_v \right\|^2 \geq 4c \log(n) - c$$
$$\mathbf{t}_v^\top K^\top Q \mathbf{t}_v = \left\| \mathbf{x}_v \right\|^2 + c \left\| \mathbf{z}_v \right\|^2 - 3c \leq 4c \log(n) - 2c$$
$$\mathbf{t}_v^\top K^\top Q \mathbf{t}_w = \langle \mathbf{x}_v, \mathbf{x}_u \rangle + c \langle \mathbf{z}_v, \mathbf{z}_u \rangle - 3c \leq 4c \log(n) - 2c$$

This shows that $\mathbf{t}_v^\top K^\top Q \mathbf{t}_u \geq \mathbf{t}_v^\top K^\top Q \mathbf{t}_v$ and $\mathbf{t}_v^\top K^\top Q \mathbf{t}_u \geq \mathbf{t}_v^\top K^\top Q \mathbf{t}_w$. In particular, $v$ will attend to its child with $\max_{u \in \mathcal{C}(v)} \langle \mathbf{x}_v, \mathbf{x}_u \rangle$, which is the correct answer to the sub-question represented by $v$. After applying the $V$ matrix and adding the residual connection, the to embedding of $v$ is equal to:

$$\tilde{\mathbf{t}}_v = \begin{pmatrix} \mathbf{x}_v \\ \mathbf{x}_u \\ \mathbf{z}_v \\ \mathbf{z}_{\mathcal{P}(v)} \\ \mathbf{w}_{I(v)} \\ \mathbf{w}_{I(v)+1} \\ 1 \end{pmatrix},$$

where $\mathbf{x}_u$ maximizes the inner product with $\mathbf{x}_v$ over its children. The first layer of the MLP doesn't change the input. This can be easily done with a ReLU network since $\sigma(z) - \sigma(-z) = z$ for every $z \in \mathbb{R}$.

The second self-attention layer will have the following weights:

$$K = \begin{pmatrix} \mathbf{0}_{2d \times 2d} & & & \\ & \mathbf{0}_{4 \log(n) \times 4 \log(n)} & I_{4 \log(n)} & \\ & \mathbf{0}_{4 \log(n) \times 4 \log(n)} & \mathbf{0}_{4 \log(n) \times 4 \log(n)} & \\ & & & 0 \end{pmatrix},$$

$$Q = \begin{pmatrix} \mathbf{0}_{2d \times 2d} & & & \\ & I_{4 \log(n)} & \mathbf{0}_{4 \log(n) \times 4 \log(n)} & \\ & \mathbf{0}_{4 \log(n) \times 4 \log(n)} & \mathbf{0}_{4 \log(n) \times 4 \log(n)} & \\ & & & 0 \end{pmatrix},$$

$$V = \begin{pmatrix} \mathbf{0}_{d \times d} & I_d & & & \\ \mathbf{0}_{d \times d} & \mathbf{0}_{d \times d} & & & \\ & & \mathbf{0}_{4 \log(n) \times 4 \log(n)} & I_{4 \log(n)} & \\ & & \mathbf{0}_{4 \log(n) \times 4 \log(n)} & \mathbf{0}_{4 \log(n) \times 4 \log(n)} & \\ & & & & I_{8d \log(n)+1} \end{pmatrix}.$$

The second MLP will again just copy the input, and we will not use the residual connection for the second layer [5].

We now turn to prove the correctness of the construction. The second self-attention layer forces each token $v$ to attend to the token $u$ with $I(u) = I(v) + 1$, namely the consecutive token in the reverse BFS order. The $V$ matrix of the second self-attention layer transform the embedding of this token to be similar to that of an embedding of a node. Suppose we constructed $k-1$ CoT tokens already, and for a node $v$, $I(v) = k$. By the ordering, this means that all the descendants of $v$ have being processed. Also, if $v$ have descendants that are non-leaves, then the first layer correctly solved the corresponding sub-question to them, and they were outputted as CoT tokens with a similar embedding to a leaf node. Hence, in the current iteration, the token corresponding to $v$ will necessarily attend the token corresponding to its child that solves the sub-question. In addition, the last token CoT token that was created will necessarily attend to the token corresponding to $v$ by the construction of the second self-attention layer. This means that the new CoT token will be the answer to the sub-question corresponding to $v$, with and embedding similar to a leaf. This is true for any node $v$, hence the last CoT token will contain the answer to the root, which is the answer to the CRQ.

---

[5]It is always possible to ignore the residual connection (if it exists) by doubling the embedding dimension. This can be done by adding zero coordinates, then use the $V$ matrix to copy the original vector intro those added coordinates, and use another linear layer from the MLP to move back the vector to the first coordinates while removing what was added from the residual connection.

# F General CRQs

In our definition of CRQs (Definition 3.1) each sub-question is defined as the argmax over inner products. There are two main reasons for this choice: (1) The argmax function aligns well with the attention layers in transformers, allowing for simple constructions, and (2) The argmax function is flexible enough to capture many natural problems as special cases, such as evaluating Boolean expressions. However, by allowing for other operations besides argmax, CRQs can be generalized to include many more tasks. We define the class of generalized CRQ problems formally below. For simplicity we only consider binary trees. (Given a CRQ over a non-binary tree, we can always convert it into an equivalent CRQ over a binary tree with at most twice as many nodes; see Appendix D.)

**Definition F.1.** *A **General Compositional Reasoning Question (GCRQ)** is a rooted binary tree $T = (V, E)$ with root $v_r \in V$, and a set of operators $f_1, \ldots, f_k : \mathbb{R}^{3d} \to \mathbb{R}^d$. Each node $v \in V$ is labeled by a vector $\mathbf{x}_v \in \Gamma^d$, and non-leaf nodes are also labeled by an operator's index $i_v \in [k]$. Here $\Gamma \subset \mathbb{R}$ is some finite vocabulary of constant size and $d \in \mathbb{N}$.*

*The **answer** to the GCRQ is defined as $\mathcal{A}(v_r)$ where the function $\mathcal{A}$ is defined recursively: For a leaf $u \in V$ we define $\mathcal{A}(u) = \mathbf{x}_u$. For a non-leaf node $v \in V$ with children $u_1, u_2$ we define:*

$$\mathcal{A}(v) = f_{i_v}(\mathbf{x}_v, \mathcal{A}(u_1), \mathcal{A}(u_2)) .$$

*We refer to every node $v \in V$ which is not a leaf as a **sub-question**.*

As a concrete example, we can represent modular arithmetic problems as a special case of GCRQs. Let $\Gamma = \{0, 1, \ldots, p - 1\}$ for some prime $p$. Let the dimension of the embedding vectors be $d = 1$. We define the operators: $f_+, f_-, f_\times, f_\div$ as:

$$f_+ = (\mathbf{x}_v, \mathbf{x}_{u_1}, \mathbf{x}_{u_2}) = \mathbf{x}_{u_1} + \mathbf{x}_{u_2}, \quad f_- = (\mathbf{x}_v, \mathbf{x}_{u_1}, \mathbf{x}_{u_2}) = \mathbf{x}_{u_1} - \mathbf{x}_{u_2},$$
$$f_\times = (\mathbf{x}_v, \mathbf{x}_{u_1}, \mathbf{x}_{u_2}) = \mathbf{x}_{u_1} \times \mathbf{x}_{u_2}, \quad f_\div = (\mathbf{x}_v, \mathbf{x}_{u_1}, \mathbf{x}_{u_2}) = \mathbf{x}_{u_1} \div \mathbf{x}_{u_2} .$$

(In this construction, the non-leaf labels $\mathbf{x}_v$ are not used.) Clearly, any arithmetic expression can be converted into a GCRQ whose tree structure matches the order of operations in the expression. Furthermore, each of the four arithmetic operators can be represented by a ReLU network. For instance, Feng et al. [2024, Lemma C.5, Theorem D.1] implements them using a lookup table, since the inputs come from a finite field.

All of our constructions can be adapted to work for GCRQs. Consider $k$ operators $f_1, \ldots, f_k : \mathbb{R}^{3d} \to \mathbb{R}^d$, where each operator can be implemented by a ReLU network. Note that the only place in the proofs of Theorem 4.1 and Theorem 6.1 we use the fact that the operator is argmax is where we use a single self-attention layer to calculate inner products. Instead, we can modify this attention layer so that each parent node attends to both its children, with the output being a concatenation of their vectors. In more detail, assume that each two siblings $u_1$ and $u_2$ in the tree are labeled by vectors $\begin{pmatrix} \mathbf{x}_{u_1} \\ \mathbf{0}_d \end{pmatrix}$ and $\begin{pmatrix} \mathbf{0}_d \\ \mathbf{x}_{u_2} \end{pmatrix}$. Then using the positional embeddings, we can force their parent to attend to both of them with the same attention score just as in the proofs of Theorem 4.1 and Theorem 6.1. We also construct the MLP so that its input will include the index of the operator, $i_v$. The MLP will simulate an "if" statement over the $k$ operators and apply the relevant one over the vectors $\mathbf{x}_v, \mathbf{x}_{u_1}, \mathbf{x}_{u_2}$. This "if" statement can be simulated by a $k$-layer ReLU network. We need to assume that $k$ is independent of $n$, which is natural since the number of possible operators (e.g. 4 in arithmetic problems) is independent of the input length.

A similar construction will work for the proof of Theorem 5.4, where here the MLP will output the last two inputs from the stack and apply the operator $i_v$. Again, it will increase the depth of the MLP by $k$. We leave exact constructions for future work.

