# OpenReview forum: "Compositional Reasoning with Transformers, RNNs, and Chain of Thought"
_NeurIPS.cc/2025/Conference — NeurIPS 2025 poster_

### Official Review · Reviewer_u4qx · 2025-06-26

**Clarity:** 3
**Significance:** 2
**Originality:** 3
**Rating:** 4
**Confidence:** 3

**Summary:**

The paper investigates the expressive power of transformers, recurrent neural networks (RNNs) and transformers with chain of thought (CoT) for Compositional Reasoning Questions --- which they formally define as tree structured hierarchical problems. They investigate the trade-offs between the number of nodes and depth of a CRQ, and the architecture requirements (number of parameters, runtime and parallel-runtime) for Transformers, RNNs and CoT.  They provide the following results:

1. The provide a sufficiency criterion on a Transformer's depth, embedding dimension for solving a CRQ (Theorem 4.1)

2. Any constant‑depth transformer, no matter how wide, fails on some CRQs unless (TC^{0} \neq NC^{1}).

3. Role of ordering of the nodes of a CRQ on the size of RNNs hidden dimensions required to solve the task.

4. They show that under correct ordering of nodes, shallow RNNs can solve CRQs

5. They show that a 2-layer transformer (with CRQ size dependent embedding sizes and precision) with CoT can solve CRQs.

**Questions:**

Q1. Do you think that your results provide any meaningful guidance to practitioners for specific applications? If yes, then such applications should be highlighted.

Q2. I am not a circuit complexity expert, but why is the result of Thm 4.3 conditional on TC$^{0} \neq NC^{1}$, when you seem to have a clear counter example (line 239)?

Q3. Are CRQs generalization/restriction of some other problems investigated in theoretical CS?

**Ethical Concerns:**

["NO or VERY MINOR ethics concerns only"]

**Final Justification:**

**Reasons for maintaining the score:**
After seeing the reviews from other reviewers and the author's rebuttal. I still believe that the paper can be interesting for the community studying expressivity of transformers. Hence, I will maintain my score.

**Reasons for not upgrading:**
The authors have attempted to address my questions and concerns regarding the naturalness and practical motivation of the investigated problem. However, even after reading their response, I remain unconvinced. I believe that the motivation behind the work is still not clear, and this concern appears to be shared by other reviewers as well.

**Limitations:**

There is no discussion of limitations. And I think some discussion of the sort should be there, as at the moment I do not see why the results are relevant to any potential application or how are they connected to other natural problems investigated in Theoretical CS.

**Paper Formatting Concerns:**

None.

**Quality:**

3

**Strengths And Weaknesses:**

S1[Originality]. A key strength of the paper is the formalization CRQs as a well-defined compositional task (although, I am slightly ambivalent on this, also check W3).

S2[Clarity and Quality]. Quite intuitive and natural results that are presented in a well-structured format. The paper clearly outlines the necessary and sufficient conditions for Transformers and RNNs to express CRQs. I think fair amount of intuition is provided for the results. I did not check the proofs in the appendix.

W3 [Significance]  Although I appreciate the CRQ formalization and the presented results. I am not sure that the paper's results actually motivate any guidance for practitioners, or explain any observed behavior in transformer design. Furthermore, the problem does not have a very natural motivation and the proof ideas (at least as discussed in the paper) do not seem to be of major impact on other areas.

---

> ### Author Rebuttal · Authors · 2025-07-30
>
> We thank the reviewer for the thorough and positive review.
>
>
> **W3 + Q1:** Thank you for raising this point. We acknowledge that we could have better framed the larger stakes of this problem and the takeaways of our results for practitioners and others working outside the field of representational capacity. Let us explain a bit more about our broader motivation for studying this problem.
>
> Transformer LLMs have proved to be surprisingly good at reasoning, especially on small problems, but still have obvious limitations. Experimental and theoretical analysis agree that they struggle, especially as the problems become larger and computationally harder. There are several competing trends or schools of thought about how to overcome these limitations:
> 1) Trust the scaling laws. Grow the size of the transformer by stacking more layers, and greater capacities will continue to emerge.
>
> 2) Devise better architectures, such as state space or hybrid models, because attention is not all you need. Improving the architecture will make it easier to learn and perform certain kinds of computations efficiently.
>
> 3) Introduce test-time scaling using chain of thought or related reasoning-style generation. Even after fixing the architecture, model size, dataset size, etc., we can squeeze out much better reasoning performance by opening up this whole new axis of the design space.
>
> Now each of these schools have received theoretical support to some extent. For instance, [1,2,3] support school #1 by showing the power of log-depth transformers. Work on state tracking tasks like recognizing regular languages and composing permutations [4] supports school #2. And recent work on the power of chain of thought [5,6] supports school #3. Each school has some claim on being the best way to improve reasoning, because they are using different theoretical rulers. For instance, if the kind of reasoning you care about most resembles state tracking, then RNNs are a natural choice and everything else will look wasteful. Whereas, if your paradigm of reasoning is dynamic programming or simulating Turing machines, then chain of thought clearly looks like the most promising path. On the third hand, for k-hop induction or graph-based reasoning problems, transformers win out.
>
> But each of these theoretical measuring sticks is limited, and may not yield very general insights. For example, we believe compositional reasoning is fundamentally important, but it doesn’t fall neatly into any of these categories. It is not a state tracking task, because it requires logarithmic writable memory. It is computationally much easier than simulating a Turing machine or doing dynamic programming. It is not parallelizable enough to be solved in O(1) rounds, but is still computationally easy compared to even one layer of quadratic attention (the same may be true for certain graph problems).
>
> So our conclusion is that none of these three schools is in the goldilocks zone. Using CRQ as a measuring stick, none is a clear winner and none is a clear loser. So we should not place all our hopes in any one of them. As table 1 shows, the best choice may depend on what resource you decide you want to optimize. Perhaps it is some blend of all three schools, or a fourth paradigm that hasn’t been developed yet. While these conclusions may not immediately change the best practices for downstream applications, we think they will help guide researchers as they address the grand challenge of improving reasoning.
>
> **Q2:** We apologize for the confusing phrasing. What we meant here is that under the assumption of $TC^0 \neq NC^1$, no transformer with a constant depth (independent of $n$) can solve all CRQs. The example we give of a balanced binary tree is a particular type of CRQ for which an input of size $n$ corresponds to a tree of depth $\log(n)$. Even limiting ourselves to these balanced binary CRQs, no transformer with constant depth can solve all CRQs of this type. (Of course, our log-depth construction from Theorem 4.1 solves *all* CRQs, including this type as a special case.)  Without the assumption that $TC^0 \neq NC^1$, our proof would break and it would be unknown whether there is a constant size transformer that solves all CRQs (although the CS community mostly believes this assumption to be true, similar to the assumption that $P \neq NP$).
>
> **Q3:** Yes, CRQs are both a restriction and a generalization of important classes of problems from theoretical CS. Most prominently, Boolean formula evaluation is a special case of CRQ, a fact that we prove in Appendix A.2 as part of the proof of Theorem 4.3. Boolean formula evaluation is fundamental in computer science and has received particular attention in circuit complexity A closely related and natural problem is evaluation of arithmetic expressions mod p. We explain the connection of this problem to CRQs in Appendix F. The advantage of studying CRQs is that it captures several problems with common mathematical structure in a single framework for a unified analysis. In addition, CRQs can be understood as lying in a special class of problems that are simultaneously $NC^1$-hard and require logarithmic space. Previous work has compared LLM architectures through the lenses of state tracking and parallelism. State tracking tasks are naturally solved by RNNs and are difficult for transformers (Merrill, Petty, Sabharwal) , while highly parallel tasks are the reverse (Stanford, Hsu, Telgarsky). CRQs occupy a theoretically interesting middle ground: they are not fully parallelizable, because they are $NC^1$-hard (Theorem 4.1), but they are also not state tracking tasks, because they require $\log(n)$-size memory that finite state machines lack (Theorem 5.5). However, CRQs are *somewhat* parallelizable and *nearly* memoryless. Perhaps this can be summarized by saying that CRQs lie in a class of problems that are “inherently compositional”.
>
> **Limitations:**
> We include a brief discussion of the limitations of our work in Section 7, but we would be happy to expand this discussion to better contextualize our results according to the advice of the reviewers.
>
> [1] Merrill and Sabharwal, “A Little Depth Goes a Long Way”, 2025
>
> [2] Stanford, Hsu, and Telgarsky, “Transformers, parallel computation, and logarithmic depth”, 2024
>
> [3] Stanford, Hsu, and Telgarsky, “Representational strengths and limitations of transformers”, 2023
>
> [4] Merrill, Petty, and Sabharwal, “The Illusion of State in State Space Models”, 2024
>
> [5] G. Feng, B. Zhang, Y. Gu, H. Ye, D. He, and L. Wang. “Towards revealing the mystery behind chain of thought: a theoretical perspective”, 2023
>
> [6] Merrill and Sabharwal, “The expressive power of transformers with chain of thought”, 2023

---

> ### Comment · Reviewer_u4qx · 2025-08-04
> **I will keep my score.**
>
> **Reasons for maintaining the score:**
> After seeing the reviews from other reviewers and the author's rebuttal. I still believe that the paper can be interesting for the community studying expressivity of transformers. Hence, I will maintain my score.
>
> **Reasons for not upgrading:**
> The authors have attempted to address my questions and concerns regarding the naturalness and practical motivation of the investigated problem. However, even after reading their response, I remain unconvinced. I believe that the motivation behind the work is still not clear, and this concern appears to be shared by other reviewers as well.

---

### Official Review · Reviewer_ruzm · 2025-07-02

**Clarity:** 2
**Significance:** 2
**Originality:** 1
**Rating:** 2
**Confidence:** 2

**Summary:**

This paper inspects RNNs, Deep Transformers, and shallow Transformers with CoT tokens in the context of Compositional Reasoning Questions (CRQ).

CRQs are defined as a tree representation of a complex question with each sub question being a child node. Each node as a vectorial representation.

The authors show that transformers with depth $L$ can solve all CRQs of depth up to $L$; RNNs with O(log n) hidden dimension and constant depth can solve all CRQs of size $n$; and transformers with O(n) CoT tokens can solve all CRQs of size $n$.

**Questions:**

In Table 1, the caption says that “_Each architecture minimizes one kind of resource at the expense of the others_”, however, in the RNN row, the minimal resource is the number of parameters (with O(log n)), similar to the CoT row;  and not the runtime ( with O(n . log n)) like what is highlighted in green in the table.
The caption is somewhat confusing, it could maybe be rephrased to something like “_each ressource has an architecture that will minimize its usage_”?

**Ethical Concerns:**

["NO or VERY MINOR ethics concerns only"]

**Final Justification:**

The paper does not provide any experimental validation of the theory proposed and authors did not express any interest in adding some during the rebutal.
In addition, when asked to clarify the problem this paper tackles by providing examples, the authors also declined to do so because it is a ``paradigmatic reasoning problem''. As such, it is personally hard for me to recommend this work.

**Limitations:**

No societal impact as this is theoretical work. The main limitation of this work is the lack of experiments and experimental analysis.

**Paper Formatting Concerns:**

ok.

**Quality:**

2

**Strengths And Weaknesses:**

## Strengths

This paper provides theoretical analysis to better understand the representation capacity of Transformers, RNNs, and Transformers with CoT tokens.

This is an important research topic to better understand and interpret Transformer architectures.


## Weaknesses

1. The paper provides theorems, proofs, and intuitions, but no experiments to validate the work. Experiments with Transformers and RNNs of a given size must be run on problems of small to large sizes to validate all the theory presented in this work.

2. CRQs are defined in the paper but no dataset is presented as examples. Presenting and releasing a dataset that follows the CRQ principles would greatly improve the quality of this work. As an example of an already existing reasoning Q&A dataset that follows a graph structure, the authors can have a look at CLUTRR by Sinha et. al, 2019 which evaluates RNNs and Transformers like this paper.

3. The distinction between Transformers and Transformers with CoT tokens is not clearly motivated. These are the exact same model architecture. The only difference is the number of input / output tokens. Given that the paper is motivated by exploring the strengths and weaknesses of neural network **architectures** for a class of problems; I would encourage further discussion why this split is necessary.

---

> ### Author Rebuttal · Authors · 2025-07-30
>
> We thank the reviewer for the thorough review.
>
> **Weaknesses**
>
> 1) As discussed in Section 7, we acknowledge that our upper bounds do not in themselves demonstrate what functions these architectures can learn, only what functions they can represent. It would be valuable for experimentalists to validate that the scaling rules predicted by our upper bounds hold in practice. However, we think that our theoretical analysis is already a valuable contribution to the literature on transformers and RNN representational capacity. First, our lower bounds apply regardless of the training methodology used in practice. Second, proving results like ours for a particular, well-defined task guides experiments by providing clean, controlled, easily-testable hypotheses. Finally, we are interested in the capabilities not just of today’s frontier models, but of architectures and problem sizes that are too large to experiment with currently. For those settings, asymptotic analysis is essential. For these reasons, approximation theory has been an important subfield of machine learning for decades.
>
> 2) The review asks us to identify a real-world example of a CRQ problem. As we prove, CRQ is a generalization of Boolean formula evaluation. All of our results — both our constructions (upper bounds) and impossibility results (lower bounds) apply even if we restrict ourselves to this task. We believe that Boolean formula evaluation is a paradigmatic reasoning problem. Just as related work has used problems like graph connectivity, recognizing regular languages, and arithmetic to examine the capabilities and limitations of reasoning models, we use Boolean formula evaluation. Moreover, the CRQ framework allows us to provide a unified analysis of other problems that share the compositional structure of Boolean formulas. Our focus is not on the outward form of the problems, which differentiates them from one another, but on the deeper properties that unite them. We see this as a strength of the paper. The CLUTRR task is a great example, and we will add a reference to it in the introduction; thank you for pointing it out to us. However, it is not a goal of this paper to devise new benchmarks with compositional structure; we believe that existing tasks like Boolean formula evaluation provide plenty of motivation for studying this kind of reasoning.
>
> 3) The review asks why we differentiate between different types of transformers, particularly why we refer to them as being different “architectures”. The reviewer suggests that these are all examples of one architecture, just with different hyperparameters. However, our paper is part of a line of work showing that hyperparameters like the number of layers and number of chain of thought tokens can have a major, qualitative impact on the model’s power. Thus, when describing a neural network architecture, it does not suffice to say “we used a transformer” or even “let’s add chain of thought prompting to our transformer”. For the types of problems we study, it is essential to specify the number of layers in the transformer and the number of chain of thought tokens we allow, relative to the size of the inputs. Our paper considers four types of transformers: (1) constant depth transformers without chain of thought, (2) constant depth transformers with logarithmic chain of thought, (3) log depth transformers without chain of thought, and (4) constant depth transformers with linear chain of thought. We show that these models are qualitatively different in that (1) and (2) cannot solve CRQs, while (3) and (4) can. Furthermore, (3) and (4) use resources very differently, as shown in Table 1. Thus, while all four of them are transformers, it is fruitful to think of them as if they were distinct architectures. We acknowledge that this use of the term “architecture” may be counterintuitive, and we will add a clarification in the introduction.
>
> **Questions:**
>
> - The review is correct in its interpretation of the table. It shows that, for each resource, there is a particular architecture that minimizes its use, and furthermore, that no single architecture is best for more than one resource. Thus, when you select one of the architectures, you are effectively prioritizing one resource at the expense of the others. In the table, it means that each architecture (row) minimizes one resource (column). For example, RNNs minimize the runtime, which is quadratic for transformers with and without CoT, and almost linear for RNNs.

---

> ### Comment · Reviewer_ruzm · 2025-08-05
> **thank you for the response**
>
> Dear authors,
>
> Thank you for taking the time to answer my concerns, however it seems like out of the three points I raised, only one has been addressed and will result in an updated introduction section.
>
> I understand that you are interested in the capabilities of architectures and problem sizes that are too large to experiment with currently, however today's architectures are already quite large and hard to experiment with. I also understand that this work will hold **for all** architecture sizes nonetheless. As such, I still think that it would be greatly beneficial to validate at least to some extent your analysis with **a** model (not necessarily the largest) on **a** dataset (could be synthetically generated) in a controlled experiment. Many theoretical findings papers will use synthetically created datasets for such controlled environments.
>
> While CRQ is a generalization of Boolean formula evaluation, and Boolean formula evaluation is a paradigmatic reasoning problem, it would be especially beneficial for the paper to have an example of such a problem for clarity purposes.
>
> Thank you for justifying the use of different transformer types. It will be good to have this justification in the main text of the paper.

---

### Official Review · Reviewer_GoH5 · 2025-07-02

**Clarity:** 3
**Significance:** 2
**Originality:** 3
**Rating:** 3
**Confidence:** 3

**Summary:**

The paper provides theoretical analysis of transformers and RNNs in solving a subset of reasoning tasks which the authors named as Compositional Reasoning Question. The analysis shows some interesting insights but the proof is only for the ideal case and without experimental support.

**Questions:**

None

**Ethical Concerns:**

["NO or VERY MINOR ethics concerns only"]

**Final Justification:**

I will maintain my rating as I still think the proposed analysis only targets a narrow set of compositional reasoning problems and experimental results are not provided.

**Quality:**

2

**Strengths And Weaknesses:**

Strength:
1. The paper is in general written clearly with easy to follow usage of math notations.
2. The paper shares some insights on the theoretical grounding of transformer and RNN's ability to solve reasoning tasks that can be structured as binary trees. The paper illustrates why CoT can allow shallower transformer to generalize for the reasoning task.
Weakness:
1. Some statements are not substantiated. e.g. Line 168 "(though the lower bounds generally hold against models that
include them)". The authors also didn't explain why they pick transformers with single attention head that uses hardmax rather than softmax (presumably to simplify the proof?).
2. The proof seems to be only for the ideal case. For example, the authors "use a different input format that encodes the tree structure directly into the positional encodings," but this in reality when the reasoning problem is natural language based and fuzzy, how well can the proof be extended to those cases?
3. While this is more of a theoretical paper, I think some experimental evidence such as results on some reasoning datasets will strength the paper more.

---

> ### Author Rebuttal · Authors · 2025-07-30
>
> We thank the reviewer for the thorough review.
>
> **Weaknesses:**
>
> - The review mentions line 168: “For simplicity, we do not use layer normalization or attention masking [when defining transformers] though the lower bounds generally hold against models that include them.” The lower bound in question is Theorem 4.3, which is a consequence of the fact that constant-depth transformers cannot solve tasks that are $NC^1$ complete (e.g., in [1]). But in [1], this fact was proven even for transformers that do have layer normalization and attention masking. Thus, our proof of Theorem 4.3 holds even without our simplifying assumption. This is a good point, and we will clarify it and add it to the theorem. The other lower bounds in Section 5 are about RNNs and do not concern the transformer architecture. If there are other claims that the reviewer thinks could be justified better, we are happy to do so.
>
>    Regarding the question of softmax versus hardmax, we indeed used the hardmax function for ease of analysis, as in previous work (e.g., [2]). However, our construction can be extended to use softmax instead of hardmax. We would need to change the guarantees to say that we can approximate the target function up to an arbitrarily small error $\epsilon > 0$ that is independent of n, rather than exactly expressing it. To give a concrete example of how this extension can be achieved, take Theorem 4.1. We use the hardmax assumption in lines 552-556 of the proof, which describes the token we want to attend to in each layer. By the definition of the softmax function, if we want 99.99% of the attention to go to the desired token, it should have an attention score that is greater by $\Omega(\log(n))$ than any of the other n tokens’ attention scores. As you can see in lines 558 - 559 this already holds for some layers, and by adjusting our constants it can be made to hold for the other layers too (line 562). In fact, since error accumulates across layers, we need $(1 - \epsilon/L)$ of the attention to go to the desired token to have final error epsilon, but since $L = \log(n)$ and epsilon is constant with respect to n, this also already holds. Thus, even a low temperature softmax suffices. This is a good point and we will add this as a remark in the final version. We should also point out that, as in other theoretical studies of attention, we don’t assume finite precision but allow the precision to grow as log n (see lines 174 and 188).
>
>
> - We acknowledge that, as in comparable theoretical work, we study an idealized task and make simplifying assumptions about components such as the input format. These simplifications only strengthen our lower bounds. **Even if** a benevolent input format is used, the models will still fail because their architectures are inherently limited. We consider the tree-aware positional encodings (and, for the RNN result, the input order) to be constructed in a preprocessing phase that assembles the input. We conjecture that these structures can be learned using the early layers of a transformer, and it would be an interesting question for future work to confirm this experimentally. Different problems can encode tree structure in various ways that each require different preprocessing steps; for instance, arithmetic formulas use parentheses, while natural language does not. Studying the general framework of CRQ problems allows us to abstract away the specific input format and focus on what we see as the more mathematically interesting part: the compositional structure itself.
>
> - As discussed in Section 7, we acknowledge that our upper bounds do not in themselves demonstrate what functions these architectures can learn, only what functions they can represent. Still, we believe that our upper bounds capture the actual abilities of these architectures to solve CRQs, especially given that findings from comparable theoretical studies like [1] and [3] do hold up in experiments. Moreover, we believe there is value in understanding the fundamental capabilities of these architectures, divorced from the training algorithm. Finally, while it would be valuable to compare the three architectures on downstream reasoning tasks or other benchmarks, we scope our paper only to CRQs. We believe CRQs provide a clean way to explain aspects of these architectures’ ability to reason at scale that are not captured in the literature.
>
>
> [1] W. Merrill and A. Sabharwal. The parallelism tradeoff: Limitations of log-precision transformers. 2023
>
> [2] Merrill, William and Ashish Sabharwal. “A Little Depth Goes a Long Way: The Expressive Power of Log-Depth Transformers,” 2025.
>
> [3] Feng, Guhao, et al. "Towards revealing the mystery behind chain of thought: a theoretical perspective," 2023.

---

> > ### Comment · Reviewer_GoH5 · 2025-08-07
> >
> > Thanks for the rebuttal. While my first concern is addressed, I still think the proposed analysis only targets a narrow set of compositional reasoning problems and experimental results are not provided. Thus I'd like to maintain my rating.

---

> ### Author Response · Authors · 2025-08-08
>
> Thank you for the additional comment.
>
> We want to emphasize that the CRQ provides a comprehensive definition of compositional reasoning problems, encompassing various problems previously studied in different works. One such problem is Boolean Formula Evaluation, which is widely studied in the theoretical CS literature as one of the central $NC^1$-complete problems.
>
> Another set of problems that CRQ covers is modular arithmetic (see Appendix F). This problem is studied extensively in "Towards revealing the mystery behind chain of thought: a theoretical perspective, Feng et al. 2024". Our work directly improves their results. For example, we show that CRQs can be solved by transformers with CoT, by generating at most $n$ tokens, while their work requires generating $\Omega(n^2)$ CoT tokens. Thus, we do not think that the formulation of CRQ is narrow since it captures in a single formulation different problems that are studied in the literature.
>
> We will emphasize the generality of the CRQ formulation in the final version.

---

### Official Review · Reviewer_71Pc · 2025-07-03

**Clarity:** 4
**Significance:** 3
**Originality:** 3
**Rating:** 5
**Confidence:** 3

**Summary:**

This paper introduces a formal problem class called Compositional Reasoning Questions (CRQs) to analyze the expressive power of different neural architectures. CRQs are problems with a tree-like structure, such as Boolean formula evaluation, that require multi-step, compositional reasoning. The authors theoretically compare the capabilities of transformers, RNNs, and transformers with chain of thought (CoT) on this task. They prove a series of complementary results showing that no single architecture is strictly best. Instead, there are fundamental trade-offs: deep transformers can solve CRQs in parallel but require depth that scales with the CRQ's tree depth; RNNs can solve them with constant depth but require a specific input ordering and a hidden state size that scales logarithmically; and constant-depth transformers with CoT can also solve them, but require a number of thought tokens that is linear in the problem size.

**Questions:**

* The constructions presented are elegant proofs of expressiveness, but they rely on very specific, pre-designed components like the tree-aware positional encodings for the transformer and the memory-rank sorting for the RNN. Do you have any intuition on whether these mechanisms, or functionally equivalent ones, could be learned by these architectures from data, or do you view these results as primarily characterizing the limits of what a pre-programmed network can do?
* Your analysis uses a hardmax attention function. How do you expect your positive results (the constructions) to change if using a standard softmax? Could a low-temperature softmax approximate the hardmax behavior, perhaps at the cost of requiring higher precision, or would it fundamentally break the constructions?

**Ethical Concerns:**

["NO or VERY MINOR ethics concerns only"]

**Final Justification:**

After reviewing the full discussion, I am maintaining my recommendation to Accept.

Resolved Issues: The authors gave solid answers to the technical questions, effectively clarifying the path from hardmax to softmax attention and the reasoning for their lower-bound assumptions.

Unresolved Issues: The main unresolved point, shared by other reviewers, is the lack of experiments. The paper proves what architectures can express, but not what they can learn.

Justification: For me, the paper's theoretical contributions are strong enough to stand on their own. The CRQ framework is a novel tool for analysis, and the rigorous comparison of architectures offers genuinely new insights into their fundamental trade-offs. In my view, the value of this foundational work is high enough to recommend acceptance.

**Limitations:**

Yes.

**Paper Formatting Concerns:**

None.

**Quality:**

4

**Strengths And Weaknesses:**

Strength

* The primary strength of this paper is its novel and insightful comparison of major neural architectures on a single, well-defined class of problems.
* The introduction of Compositional Reasoning Questions (CRQs) provides a simple yet powerful framework for studying an essential capability of language models: multi-step, hierarchical reasoning. By proving that this problem is NC¹-hard, the paper grounds the analysis in established complexity theory, providing a solid foundation for its claims.
* The theoretical analysis is rigorous and comprehensive. I appreciate that the authors provide both positive results (constructions showing how a model can solve the task) and negative results (conditional lower bounds showing what a model cannot do) for each architecture. This dual approach gives a much more complete picture of each model's capabilities and limitations. The summary of these trade-offs in Table 1 is particularly effective.
* The paper is exceptionally well-written. The motivation is clear, the CRQ problem is defined precisely, and the main results are stated upfront and then elaborated on in dedicated sections. The comparison with related work, especially Feng et al. [2024], is sharp and clearly articulates this paper's distinct contributions.
* While the proof techniques themselves (e.g., reductions from circuit complexity, communication complexity arguments) build on prior work, their application to the novel CRQ framework to draw a three-way comparison between transformers, RNNs, and CoT is highly original. The resulting insights into the architectural trade-offs feel new and important.

Weakness:
* The main weakness lies in the gap between expressiveness and practical learnability. The constructions for solving CRQs rely on highly engineered mechanisms, such as specific positional encodings that perfectly capture the tree structure or a presorted input sequence for the RNN. It is an open question whether these solutions can be learned via standard gradient-based optimization, a limitation the authors rightfully point out in their future work section.
* The analysis uses a hardmax attention mechanism and assumes finite precision, which are standard in theoretical analyses but differ from the softmax attention used in practice. A brief discussion on how these results might translate to the standard setting would have been beneficial.

Overall, the strengths far outweigh the weaknesses. This is a strong theoretical paper that makes a significant contribution to our understanding of the fundamental capabilities of different language model architectures.

---

> ### Author Rebuttal · Authors · 2025-07-30
>
> We thank the reviewer for their thorough and positive review.
>
> **Weaknesses:**
>
> - We agree with the reviewer and acknowledge that our theoretical analysis assumes a specific, benevolent format for the input. We consider the tree-aware positional encodings (and, for the RNN result, the input order) to be constructed in a preprocessing phase that assembles the input. We conjecture that these structures can be learned using the early layers of a transformer, and it would be an interesting question for future work to confirm this experimentally. Different problems can encode tree structure in various ways that each requires different preprocessing steps; for instance, arithmetic formulas use parentheses, while natural language does not. Studying the general framework of CRQ problems allows us to abstract away the specific input format and focus on what we see as the more mathematically interesting part: the compositional structure itself.
>
> - We use hardmax attention for ease of analysis, following previous work like [1]. However, all our constructions can be extended to softmax. We would need to change the guarantees to say that we can approximate the target function up to an arbitrarily small error $\epsilon > 0$ that is independent of $n$, rather than exactly expressing it. To give a concrete example of how this extension can be achieved, take Theorem 4.1. We use the hardmax assumption in lines 552-556 of the proof, which describes the token we want to attend to in each layer. By the definition of the softmax function, if we want 99.99% of the attention to go to the desired token, it should have an attention score that is greater by $\Omega(\log(n))$ than any of the other n tokens’ attention scores. As you can see in lines 558 - 559 this already holds for some layers, and by adjusting our constants, it can be made to hold for the other layers too (line 562). In fact, since error accumulates across layers, we need $(1 - \epsilon/L)$ of the attention to go to the desired token to have final error epsilon, but since $L = \log(n)$ and epsilon is constant with respect to n, this also already holds. Thus, even a low temperature softmax suffices. This is a good point and we will add this as a remark in the final version. We should also point out that, as in other theoretical studies of attention, we don’t assume finite precision but allow the precision to grow as $\log(n)$ (see lines 174 and 188).
>
> **Questions:**
>
> - Although our theoretical analysis takes these mechanisms for granted, we believe they (or something similar) can be learned. This is supported by previous works on mechanistic interpretability, e.g. [2] where the model learns addition of natural numbers as multiplication of complex numbers, and [3] where the transformers with CoT solve arithmetic problems. These are not exactly the structures we look into, but this line of work provides evidence that transformers can learn to parse and process inputs with complex structures. Further empirical study is a very interesting future direction.
>
> - See the answer above.
>
> [1] Merrill, William and Ashish Sabharwal. “A Little Depth Goes a Long Way: The Expressive Power of Log-Depth Transformers,” 2025.
>
> [2] Nanda, Neel, et al. "Progress measures for grokking via mechanistic interpretability," 2023.
>
> [3] Feng, Guhao, et al. "Towards revealing the mystery behind chain of thought: a theoretical perspective," 2023.

---

> > ### Comment · Reviewer_71Pc · 2025-08-09
> >
> > I've read the other reviews and the author rebuttal, and I'm sticking with my original score.
> > The main criticisms from the other reviewers, e.g., the lack of experiments and questions about practical relevance, are valid points . This concern mirrors the main weakness I also noted about the gap between theory and practice .
> >
> > However, this doesn't change my assessment. The authors gave solid answers to the specific technical questions . I still believe the paper's theoretical contribution is a strong one. The CRQ framework is novel, the analysis is rigorous, and the insights from the three-way architectural comparison are valuable . For me, these strengths are significant enough to outweigh the acknowledged theoretical scope of the work.

---

### Note · Authors · 2025-08-14

We thank the reviewers and AC for their constructive engagement in the discussion period. We’d like to briefly summarize the state of the discussion as we see it.

There are two main points that remain outstanding. The first is that we do not provide an experimental study or release a new dataset (as requested by reviewer ruzm). We contend that our many theoretical results — which encompass both upper and lower bounds for three architectures and directly improve on past work — are sufficient contributions for one paper. All reviewers acknowledged the value of theoretical analysis, although reviewers GoH5 and ruzm seemed to consider experimental evidence an absolute necessity. We respectfully disagree. Theoretical papers like ours sometimes do and sometimes do not contain experiments, but in either case, the main contributions are the theorems, which stand alone.

The second point was the motivation for studying CRQ problems. Reviewers asked for examples of CRQ problems which we provided. Previous impactful papers in our subfield have studied familiar algorithmic problems like arithmetic (Feng et al. 2024), recognizing regular languages (Merrill & Sabharwal 2025), and sparse averaging (Sanford Hsu Telgarsky, 2023). These tasks are chosen both because they are important in themselves, and because they exhibit complexity-theoretic properties that help us understand the strengths and limitations of various architectures. For instance, recognizing regular languages is a paradigmatic state tracking task; it captures a certain style of reasoning. We could have done the same with the task of Boolean formula evaluation. This task is ubiquitous both in practice and in complexity theory (where it is a central $NC^1$-complete problem). Additionally, as discussed in our reply to reviewer u4qx, it is a paradigm of an interesting style of reasoning: compositional reasoning. However, by defining the broader class of Compositional Reasoning Questions, we can give a unified analysis of other tasks. One example is modular arithmetic, previously studied in Feng et al. 2024; we improve their analysis in several ways, including reducing the required number of generated CoT tokens from $\Omega(n^2)$ to $n$. Another is word or logic problems with a compositional structure, as studied in practical papers like the CLUTRR dataset suggested by ruzm. We believe the CRQ framework makes our paper strictly stronger than any single paper studying a specific task under the CRQ framework.

---

### Decision · Program_Chairs · 2025-09-17

**Decision:**

Accept (poster)

**Comment:**

This paper analyses the expressive power of different neural architectures—Transformers, RNNs, and Transformers with Chain of Thought (CoT)—on a class of problems termed Compositional Reasoning Questions (CRQs).
The reviewers are in consensus about several key strengths:

The introduction of the CRQ framework to compare the expressive powers of different architectures is an interesting approach, providing a formal basis for studying compositional reasoning. The theoretical analysis offers both upper and lower bounds for each architecture and grounding the work in established complexity theory. The comparison highlights fundamental trade-offs in resource usage (e.g., depth vs. sequence length vs. hidden state size).
The paper is overall well written with a well-defined problem, a clear description of the theoretical results, and a thorough discussion with related work.

Although the reviewers primarily concerned lack of experimental validation. The theoretical contributions are strong enough to stand on their own. The work is technically solid and provides insightful comparisons that contribute to our understanding of the fundamental capabilities of the models for compositional reasoning. Therefore, despite the lack of experiments, it can lead to future theoretical investigations into the compositional reasoning abilities of deep networks. Therefore, I recommend acceptance.